# Ecophysiological adaptations shape distributions of closely related trees along a climatic moisture gradient

Duncan D. Smith [1,2,3] ✉, Mark A. Adams [2], Amanda M. Salvi[1], Christopher P. Krieg[1], Cécile Ané [1,4], Katherine A. McCulloh[1] & Thomas J. Givnish [1] ✉

Tradeoffs between the energetic benefits and costs of traits can shape species and trait distributions along environmental gradients. Here we test predictions based on such tradeoffs using survival, growth, and 50 photosynthetic, hydraulic, and allocational traits of ten *Eucalyptus* species grown in four common gardens along an 8-fold gradient in precipitation/pan evaporation ($P/E_p$) in Victoria, Australia. Phylogenetically structured tests show that most trait-environment relationships accord qualitatively with theory. Most traits appear adaptive across species within gardens (indicating fixed genetic differences) and within species across gardens (indicating plasticity). However, species from moister climates have lower stomatal conductance than others grown under the same conditions. Responses in stomatal conductance and five related traits appear to reflect greater mesophyll photosynthetic sensitivity of mesic species to lower leaf water potential. Our data support adaptive cross-over, with realized height growth of most species exceeding that of others in climates they dominate. Our findings show that pervasive physiological, hydraulic, and allocational adaptations shape the distributions of dominant *Eucalyptus* species along a subcontinental climatic moisture gradient, driven by rapid divergence in species $P/E_p$ and associated adaptations.

Physiological ecology provides insights into how traits allow organisms to survive and compete successfully under different conditions, and thus into what shapes the distributions of species and traits along ecological gradients. Other things being equal, organisms whose traits maximize realized growth under a particular set of conditions should have an advantage in competing under those conditions.

In terrestrial plants, the photosynthetic benefits of traits (i.e., carbon gain) must be weighed against the inevitable costs of associated water loss, which include greater allocation to unproductive roots or xylem, negative effects of reduced leaf water potential ($\psi_{leaf}$) on photosynthesis and water transport, and increased risks of tissue

damage or death[1–7]. Maximizing the difference between photosynthetic benefits and transpirational costs should create advantages in growth and competitive ability. Traits thus favored by competition and natural selection should vary with environmental conditions. Optimality models involving this economics of gas exchange have yielded many testable predictions for trait-environment relationships within and among species as water availability increases relative to evaporative demand and have been supported by results of global analyses of environmental gradients[8–13]. For example, increased stomatal conductance and decreased leaf thickness and reflectance all increase photosynthesis per unit leaf mass but increase

[1]Department of Botany, University of Wisconsin-Madison, Madison, WI 53706, USA. [2]Faculty of Science, Engineering, & Technology, Swinburne University of Technology, Hawthorn, VIC 3122, Australia. [3]School of Ecosystem and Forest Sciences, University of Melbourne, Creswick, VIC 3363, Australia. [4]Department of Statistics, University of Wisconsin-Madison, Madison, WI 53706, USA. ✉e-mail: ddsmith3@wisc.edu; givnish@wisc.edu

transpiration per unit leaf mass. Optimal conductance should thus increase, and leaf thickness and reflectance decrease, in more humid areas and on moister soils, where transpirational costs are lower. Our ability to predict relationships among species distributions, growth rates, traits, and environmental conditions driven by competition and adaptation is crucial to our more general capacity to predict outcomes of environmental change.

Theory predicts that, for plants with the same leaf phenology, moister conditions should favor greater leaf width[1], stomatal conductance ($g_s$, mol m$^{-2}$ s$^{-1}$)[1–3,5,7,14–18], specific leaf area (SLA, cm$^2$ g$^{-1}$)[19], leaf N concentration ($N_{mass}$, mg g$^{-1}$)[20,21], photosynthetic rate per unit leaf mass ($A_{mass}$, μmol g$^{-1}$ s$^{-1}$)[3,5,6,21] and per unit leaf area ($A_{area}$, μmol m$^{-2}$ s$^{-1}$)[5], stomatal density[22], leaf hydraulic conductance ($k_{leaf}$, mmol m$^{-2}$ s$^{-1}$ MPa$^{-1}$)[23], stem hydraulic conductivity ($K_{stem}$, g s$^{-1}$ mm$^{-1}$ MPa$^{-1}$)[7,24], [CO$_2$] in leaf internal airspaces ($c_i$, μmol mol$^{-1}$)[5,9,18], leaf water potential ($\psi_{leaf}$, MPa)[5,7], fractional allocation to leaves and stems vs. roots[1,3,5,6], ratio of vein spacing to vein depth ($dx/dy$)[24], xylem conduit diameter[25] and relative growth rate in mass (RGR, mg g$^{-1}$ day$^{-1}$)[1,6,21] and height (mm cm$^{-1}$ day$^{-1}$)[6,21]. Conversely, drier conditions should favor greater leaf thickness[19,20], vein density (VLA, mm mm$^{-2}$)[26,27], leaf N content per unit area ($N_{area}$, mg cm$^{-2}$)[14,15], stomatal pore length[22], $A_{area}/g_s$[14], leaf reflectance[4], conduit density[15,25], and wood density[15,25]. Adaptation to fire, herbivores or pathogens, and tradeoffs involving allocation to unproductive support tissue should favor relatively thicker and denser bark[28,29] in resprouting trees (and thin bark in obligate seeders[30] including study species *Eucalyptus nitens*, *E. regnans*, and *E. viminalis*), greater cuticle thickness[31], and more stems per plant[32] in drier, shorter, more frequently burnt vegetation in drier or more seasonal sites, and greater height at a given shoot mass in more crowded vegetation[33] in moister or less seasonal sites. A summary of predicted trends in plant traits with moisture supply with underlying rationales based on published optimality models and functional tradeoffs is provided in Supplementary Data 1.

The key assumption underlying optimality models—that competition and selection favor traits that maximize growth in mass or height —is rarely tested (but see Givnish & Montgomery[34]). We expect adaptive cross-over, with species having a growth advantage under

conditions like those they dominate in nature and a disadvantage elsewhere[34,35].

Here we test theoretical predictions for many traits—and the usually untested optimality assumption, involving adaptive cross-over —based on multi-year measurements of survival, growth, and 50 other photosynthetic, hydraulic, and allocational traits of ten *Eucalyptus* species. All species were grown in four common gardens on upland sites along an eightfold natural gradient in climatic moisture supply (measured by the ratio of precipitation to pan evaporation, $P/E_p$, mm mm$^{-1}$) in Victoria, Australia (Fig. 1, Tables S1, S2). Study species were stratified by subgenus (*Eucalyptus* vs. *Symphyomyrtus*) and dominance of different portions of the $P/E_p$ gradient. Species $P/E_p$ values ranged from 0.19 for *E. dumosa*, native to arid mallee, to 0.98 for *E. regnans*, native to tall wet sclerophyll forests and Earth's tallest angiosperm. Common gardens had site $P/E_p$ values ranging from 0.16 at Hattah (surrounded by native mallee) to 0.39 at Bealiba (eucalypt woodland), and 1.03 and 1.25 at Mt. Disappointment and Toolangi (tall wet sclerophyll forest; Fig. 1, Table S2). Our study gradient shows little variation in mean annual temperature across sites (11.2–16.9 °C), little seasonality in rainfall (a small dip in summer), and little variation in rainfall seasonality across sites (Fig. S1). Temperature differences across sites should have a small direct effect, but a large indirect effect via their impact on $E_p$ (together with effects of cloudiness and humidity, both coupled to precipitation $P$) and thus on relative moisture supply $P/E_p$. $P$, $E_p$, and MAT are all strongly intercorrelated across study sites, with $P/E_p$ very strongly correlated to $P^2$ alone ($r > 0.999$; Table S2).

Our approach of scoring traits and correlates of fitness in reciprocal transplants to multiple common gardens across a range of field conditions has been widely used to study local adaptation of populations within species, dating back to the classic studies of Clausen et al.[36–41]. This approach has been used less often to quantify the relative growth (and presumed competitive ability) or survival of different species along gradients and, thus, the causes of their differential distributions[34,42].

Our experimental design permits three critical tests. First, comparisons of a trait among species within a common garden allow us to identify genetic differences among species and conduct a "soft test of

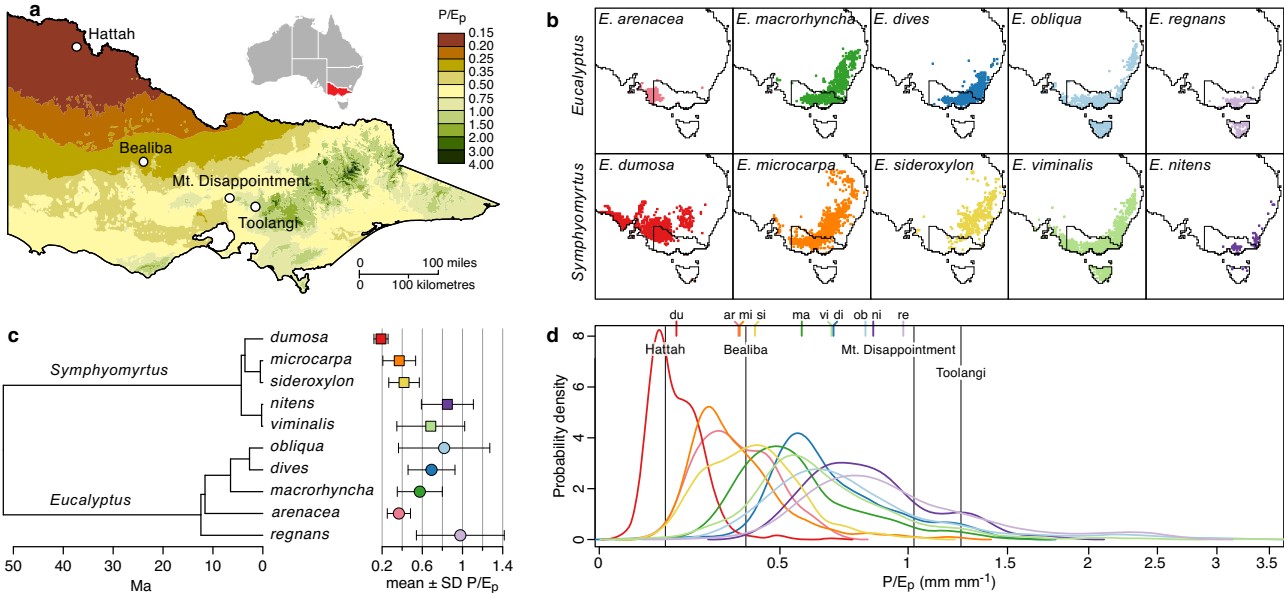

Fig. 1 | Common gardens, study species and their relationships to each other and to $P/E_p$. a Geographic gradient in $P/E_p$ in Victoria, Australia, with locations of the four common gardens; b geographic distributions of study species of *Eucalyptus*, based on $n = 56$ to 600 occurrence locations per species, stratified by subgenus (rows) and dominance of different parts of the climatic moisture gradient (columns); c estimated times of divergence and mean ± s.d. species $P/E_p$; and d probability densities of the distributions of species and sites along the $P/E_p$ gradient. Species occurrence and $P/E_p$ data are provided as a Source Data file; panels (a) and (b) are reprinted from Salvi et al. [45]. With permission from John Wiley and Sons. ©2021 The Authors. ©2021 New Phytologist Foundation.

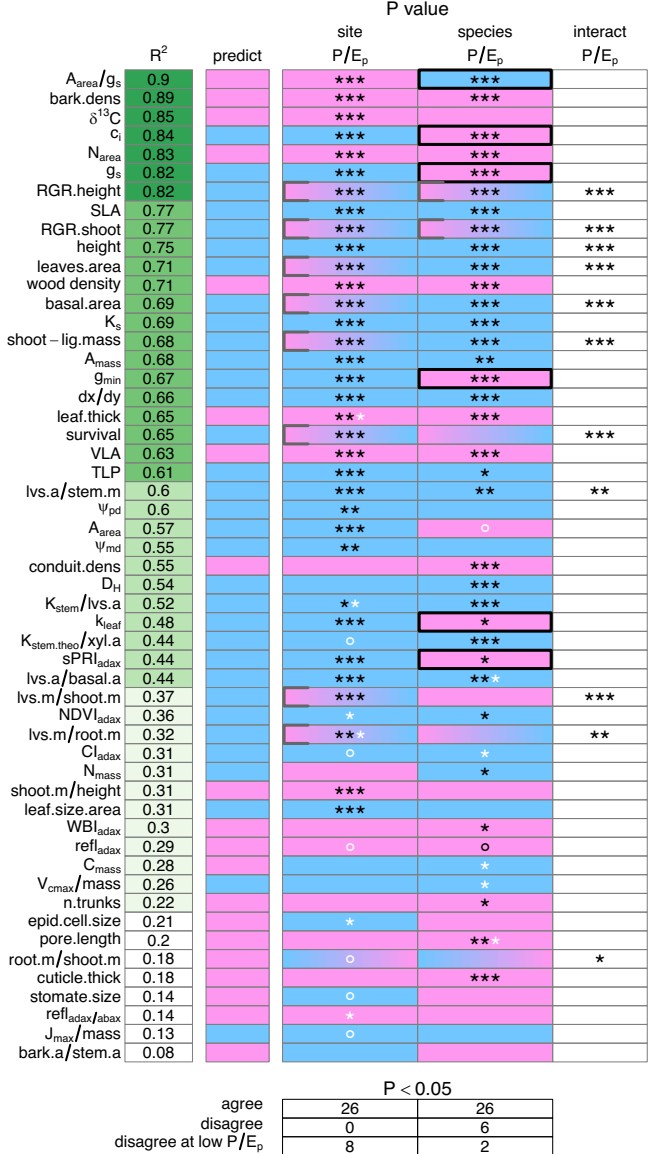

**Fig. 2 | Direction, statistical significance, and accord with predictions for trait relationships to species and site P/$E_p$ based on phylogenetic regression.** $R^2$ refers to the total two-dimensional fit. Blue indicates traits that increase with P/$E_p$; pink, traits that decrease with P/$E_p$. Color gradients indicate a shift in sign as P/$E_p$ increased left to right. Symbols ∘, *, **, *** indicate statistical significance at the 0.1, 0.05, 0.01, and 0.001 levels, corrected for multiple comparisons (white stars indicate significance without Yekutieli-Benjamini-Hochberg correction for multiple comparisons). Thick black walls identify significant relationships counter to theory; gray walls, significant relationships that accord with theory except for the driest sites or species (see also Supplementary Data 2 and Table S3). The bottom table summarizes the numbers of relationships that agree or disagree with predictions at $p < 0.05$. Source data including trait definitions are provided as a Source Data file.

adaptation"[34] by asking if the observed relationships of a trait to relative moisture supply in their home ranges (as measured by species P/$E_p$ averaged over each species' range) accord qualitatively with economic theory. Second, comparisons of a trait within single species across gardens (whose relative moisture supplies are quantified by site P/$E_p$) allow us to describe their reaction norms—patterns of plasticity—and test whether they appear to be adaptive, and whether trait responses to site P/$E_p$ (and thus, to conditions immediately experienced) are stronger or weaker than responses to species P/$E_p$ (which reflect ecological and evolutionary history). Finally, comparisons of traits across

species and gardens allow us to detect differences among species in reaction norms. Comparing reaction norms for growth permits a "hard test of overall adaptation"[34]: Do species exhibit adaptive cross-over, with growth greatest relative to others at sites with P/$E_p$ like those they dominate in nature, and with species rank shifting with site P/$E_p$? Many eucalypts show shade-intolerance—saplings are rare under continuous canopies. Hence, competition and selection should favor species with the highest realized rate of height growth, not mass growth. Our study focused on traits and survival in the seedling to sapling stages of growth as these are periods when competitive abilities are at a premium.

Here we show that most traits accord qualitatively with predicted responses to relative moisture supply; that stomatal conductance and a few allied traits show surprisingly contrasting responses to species P/$E_p$ vs. site P/$E_p$, that this result appears explicable in terms of species differences in mesophyll photosynthetic sensitivity to reduced water potential; and that most species show higher realized height growth than others at sites within their native range. Overall, our results are consistent with competition and natural selection favoring traits that maximize growth under local conditions and in such adaptations shaping species distributions.

## Results

### Trait-environment relationships accord with theory

Phylogenetically structured regressions show that most trait-environment relationships accord qualitatively with theory, with 90% of all 53 relationships within species across gardens (reflecting plasticity in response to site P/$E_p$) consistent with theory, as well as 81% of all 53 relationships across species within gardens (reflecting fixed genetic differences in response to species P/$E_p$), and weighting agreement by species and sites (Figs. 2, 3, S2, Table S3; see tallies in Supplementary Data 2). In all, 64% of trait relationships to site P/$E_p$ and 53% of trait relationships to species P/$E_p$ accord significantly or highly significantly with theory (Figs. 2, 3, S2; Supplementary Data 2; Table S3). These totals include eight traits that accord with site P/$E_p$ predictions for all species except *E. dumosa*, the species dominating the driest region; one trait accords with such predictions except for two species from the driest regions. Two traits accord with site P/$E_p$ predictions for all gardens except the driest one; three agree with such predictions except for the two driest sites. Several non-significant trait responses agreed with predictions for all sites and species or disagreed only for the driest site or species. Figure 3 shows the partial and combined effects of site and species P/$E_p$ on five exemplar traits. Among these, $A_{mass}$, SLA, and wood density increased in response to both species and site P/$E_p$ as predicted. $N_{mass}$ increased with species P/$E_p$ but showed no significant response to site P/$E_p$.

Only six of 106 trait × species or site P/$E_p$ regressions significantly contradicted predictions; five of these apparent exceptions—including stomatal conductance—are, however, explicable in terms of a more sophisticated model[5] (see Discussion). Another eight patterns are consistent with predictions except for the driest site(s) or species. Hence, almost all exceptions to predictions are predicted on more realistic grounds (and are thus not exceptions) or share a pattern in partially deviating from predictions in dry areas. Phylogenetically unstructured regressions provide very similar results, in terms of qualitative agreements with theory, levels of significance and explanatory value, and patterns of exception to predictions (Figs. S3, S4; Table S4).

### Ordination separates traits into adaptive, maladaptive, fixed, and plastic responses

A phylogenetically unstructured principal components analysis largely confirms the patterns detected by phylogenetic regression. Two PCA axes account for 58% of trait variation across sites and species, with orthogonal trait relationships implied to species vs. site P/$E_p$ (Fig. 4a).

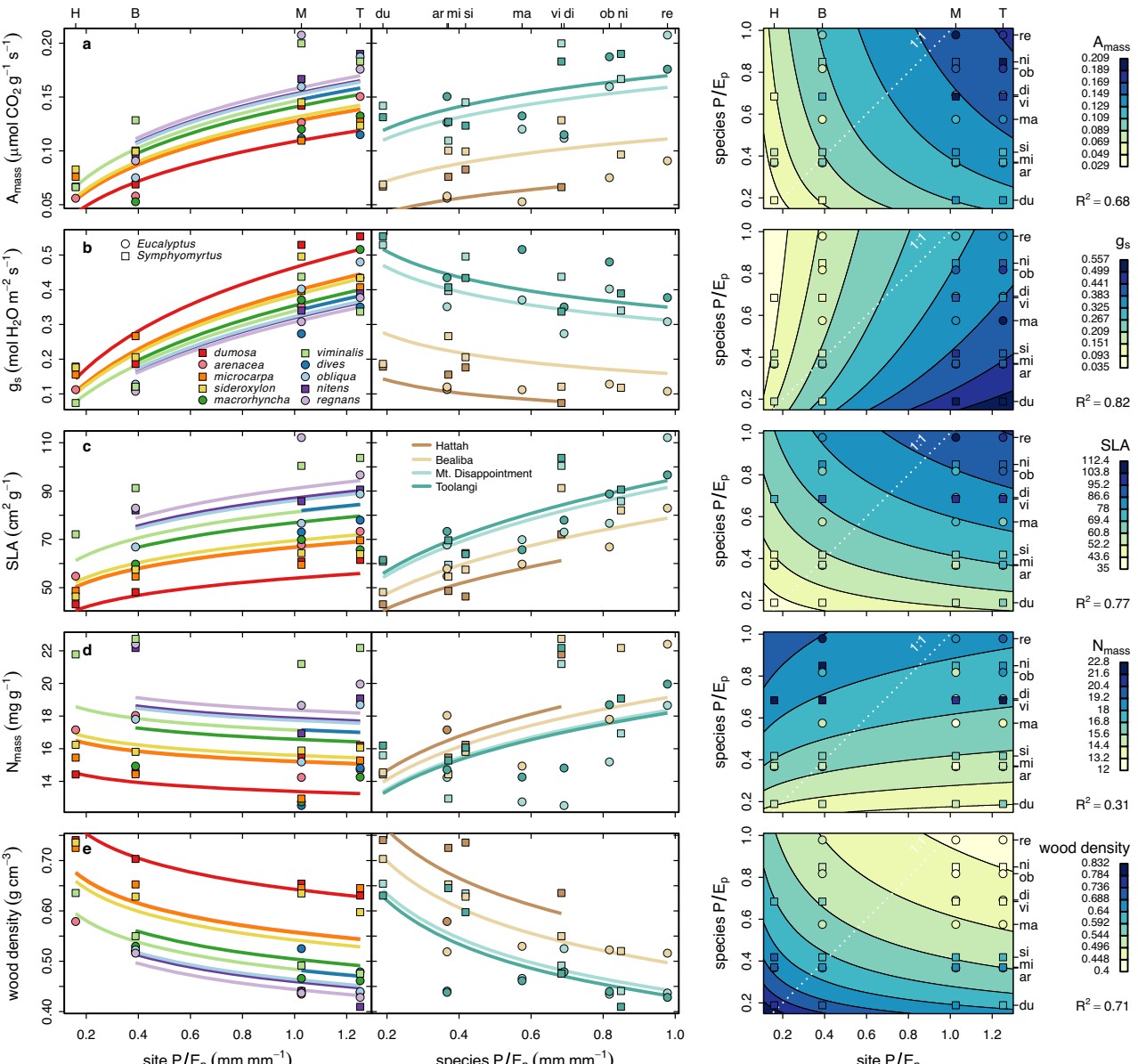

**Fig. 3 | Partial and combined effects of site $P/E_p$ and species $P/E_p$ using phylogenetic regression.** Five example traits shown are (**a**) mass-specific assimilation rate $A_{mass}$, (**b**) stomatal conductance $g_s$, (**c**) specific leaf area SLA, (**d**) nitrogen content per mass $N_{mass}$, and (**e**) wood density. For each variable, the first two graphs show the partial effects of site and species $P/E_p$, while the contour map shows the trait as a function of both parameters. Solid curves indicate regressions with $p < 0.05$. On the left, species are color-coded from driest (red) to moistest (purple) native habitats; in the middle, gardens are color-coded from driest (tan) to moistest (teal). Similar plots for all variables are shown in Fig. S2 of the online supplement. Source data are provided as a Source Data file.

Nominally, trait vector alignments relative to $P/E_p$ vectors indicate the strength of plastic responses vs. fixed differences and indicate whether trends appear adaptive (i.e., as predicted) or maladaptive. We defined eight descriptive (not prescriptive) sectors to visualize these patterns. Most traits show expected relationships to species $P/E_p$ (in the two opposing light green sectors, parallel or antiparallel to species $P/E_p$ and orthogonal to site $P/E_p$) or site $P/E_p$ (in the two light blue sectors, parallel or antiparallel to site $P/E_p$ and orthogonal to species $P/E_p$). Traits in the opposing dark green sectors show adaptive relationships to both site and species $P/E_p$. Only a few traits—stomatal conductance and related traits—show adaptive plastic responses to site $P/E_p$ (with vectors aligning near site $P/E_p$) but seemingly maladaptive fixed responses to species $P/E_p$, (with red vectors pointing opposite to that for species $P/E_p$), as seen in the top (salmon) sector of Fig. 4a. These patterns largely parallel the outcomes of the phylogenetic regressions, as shown by color-coding of vectors and terminal dots. Nominally

maladaptive patterns of plasticity (with vectors pointing opposite to expectations for site $P/E_p$) are seen in three traits in the magenta sector. Of these, only $N_{mass}$ shows maladaptive plasticity based on phylogenetic regression; $A_{area}/g_s$ shows maladaptive fixed patterns across species; and root mass/shoot mass shows nonsignificant, positive, and negative relationships to both site and species $P/E_p$ (Fig. 4a, S2). Patterns of trait-trait covariation should correspond roughly to how similar their vector directions are.

When we plot PCA axis 1 scores against species $P/E_p$ at each garden, we find that these multivariate trait summaries increase strongly with species $P/E_p$ within gardens, reflecting fixed adaptive differences in response to historically moister climates for each species over its native range (Fig. 4b). These responses are closely parallel (no statistically significant difference in slopes) among gardens, with species from moister origins exhibiting more mesic traits. Similar regressions against site $P/E_p$ for each species show that these multivariate reaction

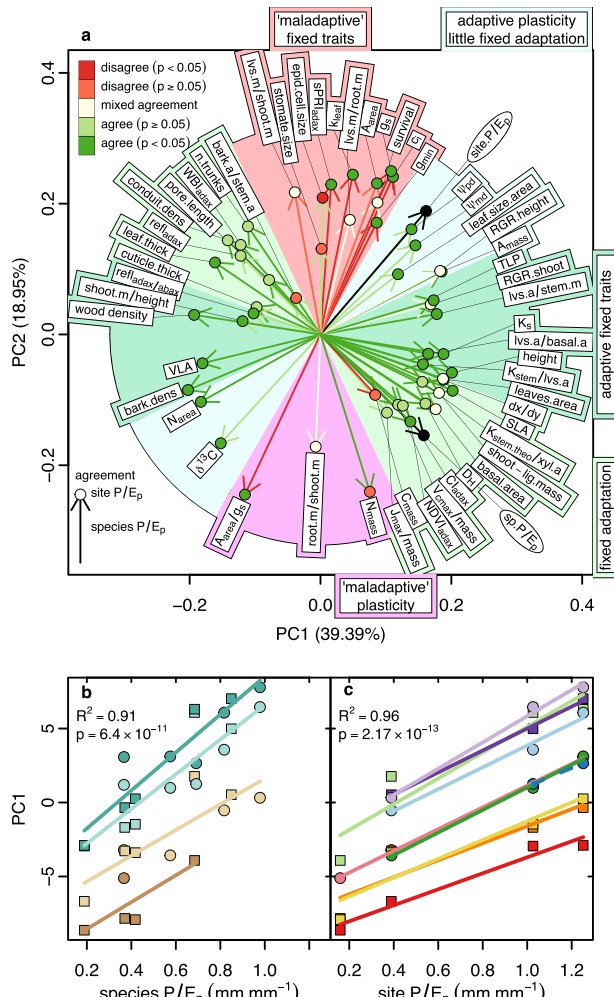

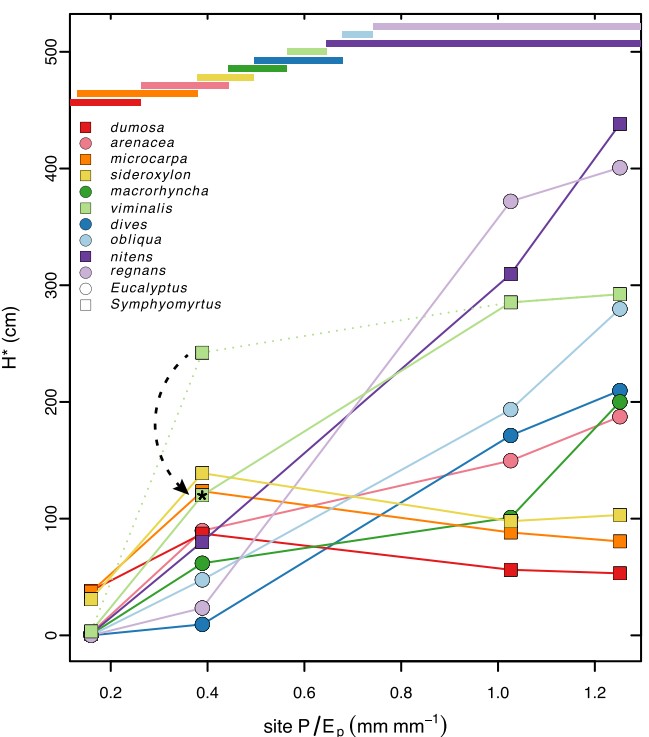

**Fig. 4 | PCA of 53 traits for all species × site combinations. a** Trait vectors in the first two PCA axes. Sectors are color-coded to indicate accord with predictions for site $P/E_p$ (blue), species $P/E_p$ (green), and both site and species $P/E_p$ (dark green); accord with predictions for site $P/E_p$ but not species $P/E_p$, nominally showing maladaptive responses to climate (salmon); and accord with predictions for species $P/E_p$ but not site $P/E_p$, nominally showing maladaptive plasticity (magenta) but also showing maladaptive fixed responses and variable responses. Vector color indicates direction and statistical significance of responses to species $P/E_p$ relative to predictions in phylogenetically structured regressions; dot color indicates the same for the response to site $P/E_p$ (see legend). Species and site $P/E_p$ were nearly perpendicular to each other when projected onto the PCA. **b, c** Multiple regression of axis 1 scores against species $P/E_p$ for each color-coded site and against site $P/E_p$ for each color-coded species. Colors follow Fig. 3. Solid lines indicate significant relationships ($p < 0.05$) with the variable on the x-axis. Lines are dashed otherwise. Slopes of individual regressions did not differ significantly. The common-slope $R^2$ and p value are given. Non-linear responses to site or species $P/E_p$ are best judged by examining regressions (Fig. S2). The multivariate trait summary PC1 increases 60% more steeply with species $P/E_p$ than species $P/E_p$, indicating that fixed differences among species in traits have a stronger effect than plasticity within species across sites. Source data including trait definitions are provided as a Source Data file.

**Fig. 5 | Realized height growth H\* (survival\*final height) for each species as a function of site $P/E_p$.** Dashed curve indicates shift in H* for *E. viminalis* at Bealiba when its survival rate is replaced with that expected (*) based on regression of survival vs. species $P/E_p$ for other species there. The observed pattern largely accords with adaptive cross-over, with species that dominate stands near each site on the $P/E_p$ gradient usually having higher H* than all others there. The two most common species as a function of $P/E_p$, based on probability densities (Fig. 1d), are indicated by the color-coded bars. The two species with the highest values of H* at each site had significantly higher H* than all others, except for *microcarpa* not differing significantly from *viminalis* at Bealiba. See text and Fig. 6 for standard errors and significant differences among species. Source data are provided as a Source Data file.

norms (patterns of plasticity) are also closely parallel, each indicating more mesic trait expressions in species when grown in moister climates (Fig. 4c). The mean ± s.d. slope for multivariate plasticity (PC1 vs site $P/E_p$) is 6.64 ± 1.28, while that for multivariate fixed traits (PC1 vs. species $P/E_p$) is 10.62 ± 2.04, indicating that plastic responses to site $P/E_p$ are only 65% as strong as the fixed responses to species $P/E_p$. Species differences in multivariate trait expression when grown under a given climate different multivariate reaction norms across a climatic gradient are required for trait differences to drive differences in species

distributions along a gradient:[34] phenotypic differences under different conditions are essential to ecological sorting based on functional differences conferred by traits.

## Realized height growth exhibits adaptive cross-over

Finally, to a large extent our study species show adaptive cross-over in realized height growth—that is, average height*survival, or H*—along the relative moisture supply gradient, with most species outperforming others at $P/E_p$ values like those they dominate in nature (Fig. 5). By contrast, we found no evidence of such adaptive cross-over in survival alone, height alone, mass, or mass*survival (Figs. S5, S6). For H* we found that *E. dumosa, microcarpa,* and *sideroxylon*—with the first, third, and fourth lowest species $P/E_p$, adapted to the driest climates—have the highest H* at the driest site, Hattah, significantly greater than those of other species there. *Eucalyptus regnans* and *nitens*—the species with the two highest species $P/E_p$—have by far the highest H* at the wettest site, Toolangi. At Mt. Disappointment, those two species significantly exceed but are approached more closely by *E. obliqua,* E. *dives,* and *E. viminalis*—the species with the third, fourth, and fifth highest $P/E_p$ (Figs. 5, 6).

At Bealiba, the second driest site, relative performances in H* initially appear to be contrary to expectations. *Eucalyptus microcarpa* and *E. sideroxylon*— the species with the third and fourth lowest $P/E_p$—have higher H* than almost every other species, as expected. But they are substantially outperformed by *E. viminalis,* a species with the next, substantially higher $P/E_p$. A close examination of the data shows,

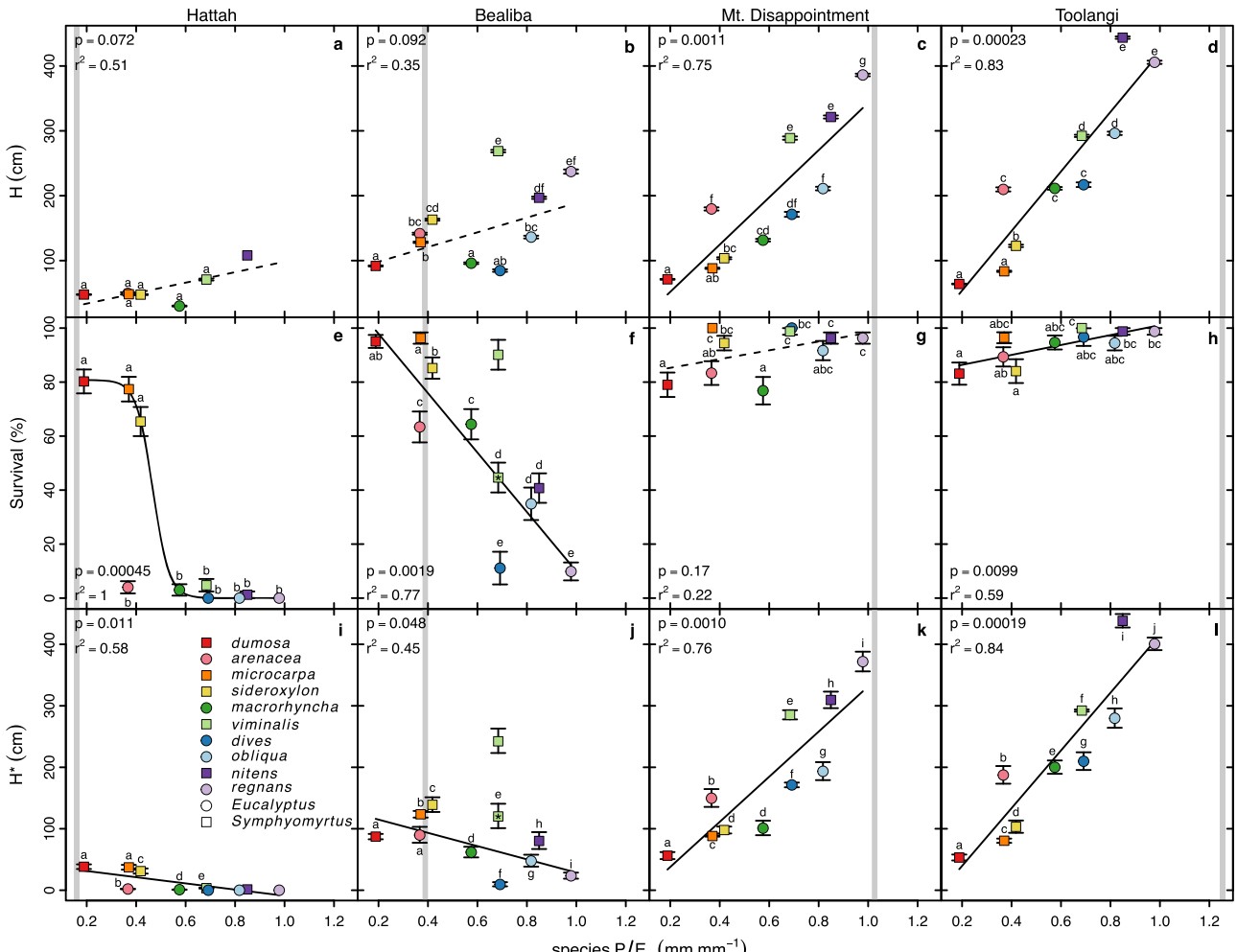

**Fig. 6 | *Eucalyptus* height growth and survival as a function of relative moisture supply.** Height growth (**a**–**d**), survival (**e**–**h**), and realized height growth (H*; **i**–**l**) of species at each garden as functions of species $P/E_p$. Vertical gray lines indicate site $P/E_p$. Data are presented as mean ± SE where the number of plants per species per site was 1–82 (height) and 27–84 (survival). See Eq. 1 for H* error calculation. Letters indicate significant pairwise differences within sites. Plotting symbols lack letters when comparisons could not be made (i.e., $n = 1$). Solid regression lines indicate

$p < 0.05$. The fit of survival at Hattah (**e**) excludes *arenacea* and was the only relationship better fit by a sigmoidal function ($s = \frac{s_{max}}{1 + \exp[\beta(a - P/E_p)]}$, where $s$ is survival, $\beta$ is a slope parameter and $a$ is the $P/E_p$ at half of maximum survival). Survival of *viminalis* at Bealiba (**f**) is shown as observed and as expected (*) based on regression of survival against species $P/E_p$ of other species there (see text). H* of *viminalis* at Bealiba (**j**) follows accordingly. Source data are provided as a Source Data file.

however, that *viminalis* had an unexpectedly high survival rate at Bealiba relative to the negative relationship to species $P/E_p$ across all other taxa at Bealiba (Fig. 6f), corresponding to unusually high soil moisture in the first months of the study (see Fig. S7). Wintertime and springtime soil moisture at Bealiba have generally declined each year of the study and height RGR of *E. viminalis* dropped substantially (see Fig. S8). When we replace the observed survival rate of *viminalis* with that expected based on the regression of survival vs. species $P/E_p$ across other species, then H* for *microcarpa* and *sideroxylon* exceeds that for *viminalis* (Figs. 5, 6f, j), consistent with adaptive cross-over and local advantage, with H* for *microcarpa* significantly greater than that for *viminalis* (Fig. 5). Linear extrapolation of H* for all species between Bealiba and Mt. Disappointment suggests that *viminalis* would have an advantage within this section of the gradient, where it is frequently a codominant (Fig. 1). Site differences in annual temperature and rainfall during the study period closely match those during the 1970–2000 reference period (Figs. S9, S10). All sites were slightly warmer than historically during the first two years of the experiment, and average to slightly cooler in the later years. For the first two years, annual rainfall was mostly somewhat below the reference period (Fig. S10). However, December 2018 was especially

rainy at Bealiba, accounting in part for the high soil moisture there early in the experiment (see above).

The patterns observed in H* reflect complex relationships of height growth and survival to species and site $P/E_p$, as shown in Fig. 6, where site $P/E_p$ is indicated as the vertical gray line on these plots for each garden. First, the height of survivors increased with species $P/E_p$ at each site, and the steepness and maximum elevation of that relationship increased with site $P/E_p$ (Fig. 6, first row). But survival declined sharply with species $P/E_p$ at the driest site; then fell less sharply, increased slightly, and increased steeply at progressively moister sites (Fig. 6, second row). The resulting pattern in H* supports adaptive crossover, with the relatively fastest growing species in each garden dominating natural sites with similar $P/E_p$, once the anomalously high survival rate of *viminalis* at Bealiba is "corrected" (Fig. 6, third row).

## Discussion

Many traits showed significant to highly significant responses to site and species $P/E_p$ in accord with theory. Especially strong, consistent, and predicted positive responses to both site and species $P/E_p$ were observed in $c_i$, SLA, height, $K_{stem}$, $A_{mass}$, $dx/dy$, turgor loss point (TLP), leaf mass/stem mass, and leaf area/basal area. Especially strong,

consistent, and predicted negative responses were seen in bark density, $N_{area}$, wood density, leaf thickness, and VLA (Fig. 2).

No trait showed a plastic response to site $P/E_p$ that significantly contradicted theory. Six traits—$g_s$ and $g_{min}$ (daytime and minimum leaf stomatal conductance), $A_{area}/g_s$ (ratio of area-based photosynthesis to stomatal conductance), $c_i$ (carbon dioxide concentration within leaf internal air spaces), $k_{leaf}$ (leaf hydraulic conductance), and $sPRI_{adaxial}$ (photochemical reflectance index on the adaxial leaf surface)—significantly contradicted theory for species $P/E_p$ (Figs. 2, 4a). These traits showed countergradient variation relative to site $P/E_p$ whereas most other traits show cogradient variation[43]. Five of these traits are tied mathematically ($A_{area}/g_s$, $c_i$, $g_s$) or adaptively ($g_{min}$, $k_{leaf}$) to stomatal conductance $g_s$, which most optimality models[1,2,7,14–18] predict should increase with moisture supply, not decrease as seen in response to species $P/E_p$ within gardens (Fig. 3e). The decreases in $c_i$ and $k_{leaf}$—and the increase in $A_{area}/g_s$—can be seen as following from the unexpected decrease in $g_s$ with adaptation to the historically greater moisture supply reflected in species $P/E_p$. The decrease in $k_{leaf}$ with species $P/E_p$ conflicts with the observed increase in VLA. Previous models for $k_{leaf}$[23] and VLA[26,27] should be re-examined to integrate the divergent effects of moisture supply and evaporative demand along the $P/E_p$ gradient. In thick-leaved eucalypts, vein density should increase in areas with low $P/E_p$—and high vapor pressure deficit—to replace high episodic rates of water loss; this may account for the tendency toward greater VLA in eucalypts in drier habitats[44].

## Mesophyll photosynthetic sensitivity may account for unpredicted patterns in $g_s$ and related traits

Unexpected patterns in $g_s$ and related traits in response to species $P/E_p$ but not site $P/E_p$ may reflect the greater decline in photosynthetic capacity at a cellular level with decreases in leaf water potentials in mesic-adapted species, which was recently demonstrated in a glasshouse study of the *Eucalyptus* taxa studied here[45]. Species from drier climates show reduced photosynthetic sensitivity to reductions in $\psi_{leaf}$. This reduced sensitivity (termed mesophyll photosynthetic sensitivity, MPS) comes at a cost of reduced maximum photosynthetic rates at full hydration[45]. Species with greater MPS should have reduced stomatal conductance based on the model advanced by Givnish[5], which could account for the unexpected patterns seen in $g_s$ and related traits within sites. Increases in $g_s$ within species toward moister sites accord with that and other models. As predicted, *Eucalyptus* species from more mesic climates operate in a narrower range of $g_s$ and $\psi_{leaf}$ than those from drier climates[46]. The unexpected increase in $sPRI$ with species $P/E_p$ within sites might reflect greater investment in the xanthophyll/zeaxanthin cycle to increase photoprotection in mesic species with higher MPS.

Several traits—like $g_s$, $\delta^{13}C$, $\psi_{md}$, $\psi_{pd}$, and $c_i$ (Figs. 3, S2)—show much stronger relationships to site than species $P/E_p$, indicating a relatively large effect of plasticity. Many of these traits are known to be plastic over short time frames and have limited costs of adjustment (e.g., transfer of osmoticum into guard cells, passive shifts in response to $\psi_{soil}$ and vapor pressure deficit, and passive responses to decreases in $g_s$), so that their plasticity in response to spatial differences in $P/E_p$ makes sense. Other traits—like $N_{mass}$, conduit diameter, cuticle thickness, various measures of leaf reflectivity, $K_{stem}$, and wood density (Fig. 3, S2)—show strong relationships to species $P/E_p$, indicating a large effect of fixed differences among species and limited plasticity. Several of these traits have high costs of adjustment, requiring construction of new leaves or conduits, or substantial nutrient translocation, so their limited plasticity with site $P/E_p$ also makes sense, with optimal levels presumably set by long-term averages of moisture supply (and other factors) in different portions of the climatic gradient. Most traits, however, like $A_{mass}$ and SLA, show strong relationships and apparent adaptation to both site and species $P/E_p$ (Fig. 3a, c, e).

Shoot mass per unit height is an important but often overlooked parameter. It declines with both site and species $P/E_p$ (Fig. S2), presumably due to selection for wide xylem elements and light wood under moist conditions, and for single slender boles and narrow crowns to enhance height growth under crowded conditions. *E. regnans* produced 30–70% less biomass than *E. nitens* at Mt. Disappointment and Toolangi (Fig. S5d), but it had only a small height disadvantage vs. the mechanically less efficient *nitens* (Fig. S5b). *E. regnans* and *nitens* have similar wood density, so plant architecture—more slender trunks, leading to markedly smaller basal areas (Fig. S2)—is what allows *regnans* to almost equal *nitens* in height growth. *Picea mariana* similarly outgrows *Larix laricina* in mass over a nutrient-supply gradient, but *Larix* gains an advantage in height growth under mineral-rich conditions due to its slenderer trunk and branches[20].

Only H* showed strong evidence of adaptive crossover (Fig. 5, S5). Presumably this reflects the importance of height growth per se (not mass growth) for competition in highly shade-intolerant eucalypts, especially at moister sites, as well as the importance of the ability to survive drought, especially at drier sites. This interpretation is supported by the prompt mortality of mesic-adapted species at Hattah, when seedlings were too small to interact, and the more gradual mortality of xeric-adapted species at moister sites as plants became big enough to interact and overtop each other (Fig. S11). Survival in large numbers *and* height growth relative to competitors are essential for competitive success.

A few species (*Eucalyptus arenacea, E. macrorhyncha, E. dives, E. obliqua*) did not place in the top two for H* in any garden (Fig. 5). Our study would have benefitted from one more common garden where $P/E_p$ ~ 0.65–0.8—e.g., near Malmsbury, VIC—where the latter three species usually dominate and where adaptive cross-over would likely occur. However, the fact that none of the four species whose relative abundance peaks at $P/E_p$ ~ 0.65–0.8 shows an advantage in height growth outside this range at any of our sites also supports our predictions. *E. arenacea*, which grows naturally at low $P/E_p$ (like Hattah) and is most commonly a dominant species on deep acidic sands, may have been inhibited by high soil pH at Hattah. Closely related *E. baxteri* becomes stunted on such alkaline sands[47,48]. Post-establishment fire—absent in our study—might have favored resprouting species with rough bark (e.g., *macrorhyncha, dives, obliqua*) over faster-growing, often non-resprouting competitors with smooth bark (e.g., *viminalis, nitens, regnans*). Herbivores and pathogens may also play an important role in determining growth and survival in some contexts. We observed conspicuous fungal lesions on *E. dumosa* leaves, native to the dry end of the $P/E_p$ gradient, when grown at Mt. Disappointment and Toolangi, the wet-end sites, where it had low survival and low growth in height and especially mass (Figs. 6, S6).

Overall, our findings support a pervasive tie among species distributions, relative growth rates, traits, and environmental conditions, driven by growth, competition, and adaptive cross-over. Results are consistent with adaptation-determining species distributions, and with most predictions of optimality models concerning how many different aspects of plant morphology, physiology, hydraulics, and allocation should vary to maximize growth at different points along an environmental gradient. Ecological sorting, evolution of adaptations to different conditions, and widespread adaptive crossover should favor different suites of traits that maximize realized height growth under different conditions and result in different species distributions (Figs. 5, S2).

Our demonstration of the adaptive value of maximizing height growth under all conditions —including the dry end of our climatic gradient—contradicts the widely cited CSR theory[49], which assumes that resource-limited habitats do not favor plants with maximal growth. This crucial assumption is based on glasshouse experiments, with no evidence that glasshouse resource levels are like those where the species in question interact in nature. Our data show that the

species that dominate different parts of the climatic moisture gradient are those with the greatest growth rates relative to others there, and that traits vary along gradients in a way consistent with optimality. Our measure of realized height growth incorporates two key components of fitness for trees—survival and height growth—which are likely to be negatively correlated when comparing a species' growth under productive conditions with its survival under unproductive conditions, even though both increase with productivity[50–52]. However, we have not yet shown that species achieve optimal trait values. That would require calculating optimal trait values (e.g., $g_s$) as a function of conditions and other traits (e.g., MPS, root hydraulic conductance) based on quantitative models and then comparing those values with observations. We note that species distributions must often be limited by maladaptive plasticity or genetic differentiation—on species failing to acquire the traits needed to maximize growth relative to others at some points along gradients. Maladaptive plasticity appears to help set distributions of Hawaiian lobeliads along light gradients[34].

The evolutionary tempo of divergence in climatic distribution and adaptation in *Eucalyptus* is remarkably rapid. In just 4.3 million years, *Eucalyptus dumosa* and *E. nitens* diverged by 4.5-fold in species $P/E_p$ (84% of the maximum seen across our study taxa in 52 My), spanning 93% of the PC1 range at Toolangi, and 77% of total PC1 range (Fig. 4). The rate of proportional divergence in species $P/E_p$ for *dumosa-nitens* of 11.4% $My^{-1}$ approaches the 17.3% $My^{-1}$ rate of evolutionary change in light regime seen in Hawaiian lobeliads[34] and exceeds the fastest known rate of sustained morphological change in plants— the increase in flower diameter by 9.5% $My^{-1}$ in *Rafflesia*[53].

Rapid divergence in climatic distributions and associated adaptations in eucalypt species correspond to extreme aridification in Australia starting 3 Mya, following a long period of drying and increased seasonality 25–10 Mya[54,55]. *Eucalyptus* began diversifying 52 Mya under warm wet conditions in the Eocene but did not become dominant until sclerophyll vegetation spread under drier conditions starting 25 Mya; most species did not evolve until arid vegetation appeared in the Plio-Pleistocene[56]. More arid-adapted subgenus *Symphyomyrtus*—including 79% of *Eucalyptus* species and the great majority of those in mallee—began rapid species diversification ~5 Mya, with the highest diversification rates in the last million years[56]. Selection should have favored traits adapted to drier conditions in species evolving in drier areas, with ecological sorting of existing species—based on context- and trait-dependent survival and competitive ability—then shaping the distributions of species with different traits along gradients of $P/E_p$. Genomic scans of populations of *Eucalyptus tricarpa* (closely related to *E. sideroxylon*) along aridity gradients in southeastern Australia suggest that adaptations to moisture supply have occurred across the genome: 73 of 94 loci showing significant deviations among sites also have significant correlations with site $P/E_p$[57]. Evidence for selection across the genome associated with relative moisture supply—and correlations with specific traits and genes—should now be sought across species with different distributions along the climatic moisture gradient. Trait responses to species and site $P/E_p$ might also be used to improve earth systems models to predict shifts in production, hydrology, and vegetation caused by climate change.

This investigation contributes to understanding how adaptations may drive differences in species distributions along an environmental gradient, combining tests of optimality theory with measurements of trait expression and plant growth over several years at multiple common gardens on that gradient, and documenting patterns in trait expression and plasticity, apparent adaptation, and adaptive crossover in realized growth that are largely consistent with theory.

Many studies have documented differences in species distribution along gradients, and several experiments have demonstrated the role of physiological tolerance and biotic interactions (e.g., competition, predation, multi-trophic interactions) in setting those distributions[58–61]. However, very few studies have gone beyond trait-

environment correlations to implicate specific traits or groups of traits as drivers of growth, survival, and species distributions along a gradient[34,61,62]. Using field experiments to examine 50 traits, survival, and height and mass growth in ten closely related species as functions of their native range and position of common gardens along a climatic moisture gradient, we identified species differences in trait expression and reaction norms that are qualitatively consistent with models of adaptation to maximize realized height growth at different points along that gradient. We showed that species differed from each other in these traits when grown under the same conditions, and that multivariate trait expression is tightly tied to species distributions. Finally, we found adaptive cross-over in realized height growth that is consistent with species distributions being set by different reaction norms for growth. Our inference—yet to be demonstrated—is that species differences in adaptation of traits/trait groups to different levels of moisture supply are what drives adaptive crossover. More studies need to conduct such "hard tests of adaptation"[34] —tied to differences in growth and survival—across groups of closely related species using multiple common gardens if we are to bridge plant physiological ecology, community ecology, and evolutionary biology in the most effective fashion.

## Methods

### Species selection and climatic distribution

We selected five species each from *Eucalyptus* subg. *Eucalyptus* and subg. *Symphyomyrtus*, stratified by the portions of the Victorian $P/E_p$ gradient they dominate, from mallee to eucalypt woodland, eucalypt forest, tall wet sclerophyll forest, and temperate rainforest[45,63]. *Eucalyptus* dominates most of this region, where its species are evergreen and shade-intolerant, rainfall is essentially aseasonal, and latitude spans only 3°. We downloaded species occurrences from *Atlas of Living Australia* (www.ala.org.au, 19 May 2018), rounding locations to the nearest 0.25° to match resolution for pan evaporation $E_p$. We removed duplicate locations and outliers (e.g., botanical gardens). For each location, we extracted mean annual precipitation $P$ from Worldclim[64] and $E_p$ from http://www.bom.gov.au/web01/ncc/www/climatology/evaporation/evapan.zip to calculate mean ± s.d. of $P$ and $P/E_p$ across the range of each species.

### Common gardens

We established common gardens in Victorian state forests at Hattah, Bealiba, Mt. Disappointment, and Toolangi, spanning an eightfold range in $P/E_p$ from 0.16 to 1.25 (Fig. 1, Table S2). Gardens were 0.25-ha upland plots in recently logged or previously cleared land, unshaded by surrounding vegetation, and fenced to exclude herbivores such as kangaroos, wallabies, wombats, and rabbits. The ground was covered with landscaping cloth (Permathene, Campbelltown NSW) to control weeds. At each site, we installed a GWRS100 weather station (Campbell Scientific, Logan UT) with sensors to log temperature, humidity, windspeed, rainfall, and volumetric water content.

Seeds for each species from 1 to 3 collections within native ranges were mixed, germinated, raised in soilless medium by Australian Native Farm Forestry (Cobram East, VIC), and planted in June 2018. We planted 81–254 individuals per species at each garden, based on variation in germination and survival. We divided seedlings for each species into enneads (3 × 3 arrays, with plants 50 cm apart), and planted these arrays randomly across three blocks, each 702–756 seedlings in all. At the two drier sites we added 1.6 mm of water at planting, and the same amount again at the driest site in July 2018. A pilot study with only 16% as many seedlings per species, without *E. dives* and *E. sideroxylon*, was planted at each site in June 2017. Intense drought required the addition of supplemental water soon after planting. Given the small sample sizes per species in this planting, we relied solely on the main experiment for most measurements, except for a few anatomical parameters measured in both the pilot and main studies (Fig. S12).

## Data collection

**Survival, height, biomass.** We censused and harvested plants from each garden three times, at 37–117 weeks after planting. We conducted each harvest as synchronously as possible (5–15 weeks) across gardens. We tallied all plants as alive or dead, and measured heights of the live plants. For the first harvest, we studied plants in the first ennead rows; the second ennead rows in the second harvest, and so on. We selected 7–8 focal plants per species stratified by height across the range in that species and recorded the number and basal diameter(s) of their basal shoots. After the first and second harvests, all unselected plants in the study rows were culled to provide more space for the remaining plants. Survival was also tallied on 3–5 additional dates outside the harvest campaigns (Fig. S12).

After measuring other traits on the focal plants (see below), we harvested them, cutting below the lignotuber if present. For the first harvest, we bagged entire plant tops, transported these to the lab to separate leaf, stem, and lignotuber tissue, collected a trunk tissue sample to determine wood density, and weighed all material after drying at 70 °C. For the remaining harvests, we used an allometric approach, estimating branch masses from their basal diameters and measuring bole mass and stem, bark, and wood densities of branches and trunks. We also estimated root mass at the time of the first harvest using excavation and allometry (see below).

We quantified % survival for each species × site × harvest combination. Relative growth rate of aboveground biomass (*RGR*, mg g⁻¹ day⁻¹) and height (*RGR*ₕₑᵢght, mm cm⁻¹ day⁻¹) was calculated as the slope of the regression of the logarithm of the geometric mean biomass (or height) at each harvest against days since planting.

**Leaf morphological traits and reflectance.** Leaf area, thickness, SLA, stomatal size and pore length, epidermal cell sizes, VLA, and spectral properties were measured on the youngest mature leaf from each focal individual. We measured leaf thickness with a dial micrometer (B.C. Ames, Waltham MA); SLA, using the area and dry mass of leaf punches. We cleared dried leaves[65] and then determined VLA in 2 mm² areas, and stomatal and epidermal dimensions from 4 to 5 cells or cell pairs. We measured spectral reflectances in the field using a UniSpec spectral analysis system (PP Systems, Amesbury MA) and then used these to calculate the normalized difference vegetation index (*NDVI*), water band index (*WBI*, a measure of water content), photochemical reflectance index (*sPRI*, a measure of plant stress reflected in the xanthophyll-zeaxanthin cycle), chlorophyll index (*CI*), and total reflectance across the visible spectrum[66]. We determined cuticle thickness and the ratio of intervein distance to vein distance to epidermis (*dx/dy*)[23] from leaf punches fixed in alcohol, embedded in LR White, sectioned to 5 µm thickness with a rotary microtome, and stained with Safranin O for microscopic measurement. All image analyses were performed in ImageJ v1.53k[67].

**Gas exchange.** During the first two harvests, we measured $A_{area}$, $A_{mass}$, and $g_s$ until they stabilized on one fully expanded leaf on each focal plant using an LI-6400XT system (LI-COR, Lincoln, NE). Most data were gathered between 09:00 and 13:00 when conditions were conducive to photosynthesis. Measurements were made at 400 ppm $CO_2$, 1700 µmol m⁻² s⁻¹ PAR, and ambient humidity except when scrubbing was needed to avoid high-humidity problems. We set the block temperature to 25 °C if leaf temperature exceeded this. $A$-$c_i$ curves were measured on additional plants to estimate $J_{max}$ and $V_{cmax}$ (see Salvi et al. [45]).

To measure minimum leaf conductance ($g_{min}$), we bagged five shoots per site × species and transported them under refrigeration to the lab during the third harvest. We rehydrated one excised leaf per shoot in distilled water overnight, measured its area, and hung it in a dark chamber over a saturated NaCl + sucrose solution used to control humidity[68], using fans to reduce boundary-layer resistance. We weighed leaves every ca. 2 h and measured the fluorescence ratio $F_v/F_m$ with a Mini PAM II (Walz, Effeltrich, Germany) every other weighing. Leaf mass typically declined rapidly and then more slowly and quasi-linearly as conductance reached or approached its minimum. We calculated conductance between each weighing using masses, humidity, temperature recorded in the chamber, and elevation-corrected barometric pressure from a nearby weather station[69]. We recorded $g_{min}$ as the average of sequential points in the quasi-linear phase with the lowest conductance and $F_v/F_m$ above 0.7. Chamber darkness was not well controlled when measuring samples from Hattah so measurements were repeated one year later using the same cohort of plants and the same sample size.

**Turgor loss point (TLP).** We collected leaves from seven plants per species × site combination during the third harvest (July–November 2020) and March–April 2021. We used the osmometry method[70], employing 12 chamber psychrometers (75-3VC, JRD Merrill, Logan UT). We calibrated chambers before the experiment using two salt-solution standards and temperature-corrected water potential calculations[71].

**k_leaf.** We measured leaf hydraulic conductance on one leaf per plant from five focal plants during the first harvest (March–April 2019), using the in vivo method[72]. We calculated $k_{leaf}$ as $T/(\psi_{stem} - \psi_{leaf})$, where $T$ is transpiration determined with an LI-6400XT system and $\psi_{stem} - \psi_{leaf}$ is the xylem pressure difference between adjacent bagged and unbagged leaves, determined using a pressure chamber (PMS, Albany OR). We excluded $k_{leaf}$ measurements when the pressure difference was small (<ca. 0.3 MPa) as this likely reflected continued transpiration by the bagged leaf or a low unbagged transpiration rate.

**Water potential.** During the first harvest (March–April 2019) we used a pressure chamber to measure predawn water potential on five of the focal plants per species. For midday water potential, we measured $\psi_{leaf}$ during $k_{leaf}$ measurements (see above). COVID-19 lockdowns, equipment theft, and the time and expense of making repeated visits along a 430-km study transect made it difficult to characterize $\psi_{leaf}$ more frequently.

**K_stem and xylem anatomy.** During October–December 2019 (68–73 weeks after planting), we placed one branch from each of five individuals per species per site in a moistened plastic bag and transported them to the lab. From each branch we excised an 11-cm segment, infiltrated this under partial vacuum in distilled water overnight to remove emboli, trimmed the ends with a razor blade and connected to a flow meter[73] filled with 20 mM KCl solution filtered to 0.2 µm to measure at least three steady-state flow rates at two or more pressures between ca. 0.4 and 4.9 kPa. Stem hydraulic conductivity ($K_{stem}$) is the slope of the regression of flow rate (g s⁻¹) vs pressure gradient (MPa mm⁻¹), which we then standardized to 20 °C and divided by cross-sectional xylem area (XA) or leaf area. $K_{stem}$ is not completely standardized by XA[74], so variation in xylem area can bias results. To avoid this, we scaled $K_{stem}$ to the median xylem area using the regression ln ($K_{stem}/XA$) ~ ln (XA) + *site* for each species where *site* is a factor.

We dried stems and then later rehydrated them and obtained sections with a sliding microtome. We stained the sections with toluidine blue to measure conduit density and hydraulically weighted conduit diameter: $D_h = 1/n(\sum_{i=1}^{n} D_i^4)^{1/4}$, where $D_i$ is the individual vessel diameter. From these measurements, we calculated the stem theoretical conductivity, $K_{stem,theo}$, using Poiseuille's Law and assuming no embolisms.

**Root biomass.** In the preliminary planting, we harvested roots at Toolangi and Hattah 61–62 weeks after planting. We excavated roots as completely as possible, cut them at branching points, measured proximal and distal diameters, dried and weighed them. When root segments had distal diameters larger than proximal diameters of other

segments, we assumed breakage and estimated the missing mass from the allometry of mass vs proximal diameter in other segments, working from smallest to largest until all roots were complete. For the main planting we harvested roots of all focal individuals. We measured all distal diameters and used the preliminary allometry to predict missing masses. We used the Hattah allometry for Hattah and Bealiba plants, and the Toolangi allometry for Mt. Disappointment and Toolangi plants.

### Analyses

**Standardized ratios.** Some traits are useful to be analyzed as ratios (e.g., leaf mass per stem mass). However, many such traits have allometric relationships, making their ratio a function of size. Consequently, comparison of simple trait ratios confounds size differences with traits of interest. To address this issue, we fit each such trait pair ($T_1$ and $T_2$) to the equation $\ln (T_1) = a_{site,species} + \beta \ln (T_2)$ to find the common slope $\beta$, and then calculate the standardized ratio as $\ln (T_1) - \beta \ln (T_2)$, which is effectively the intercept of every point when given the common slope. $\beta$ values in Source Data.

**Phylogenetic regressions.** We used phylogenetically structured regression[75] to assess how all 53 traits responded to site and/or species $P/E_p$ while taking species relationships into account. We used the time-calibrated phylogeny[45] for our taxa derived from a broader study of >700 eucalypt species[56]. We analyzed trait means for each species × site combination, excluding combinations with very low or zero survival and thus unmeasurable traits (e.g., *E. regnans* at Hattah, where all plants died). Preliminary analyses indicated that traits in natural stands often vary logarithmically with $P/E_p$ along the Victorian climatic gradient[21]. We thus log-transformed site and species $P/E_p$ and then tested additive and interactive effects between these with and without log- or square root-transforming the trait to normalize. We chose the best model based on AIC values and parsimony. We used this model as the basis for a phylogenetic generalized linear mixed model (*phyr* package[76]), in which we added four random effects: site, species, species with phylogeny, and species with phylogeny nested within site (phylogenetic attraction). From the full model we used backward selection to find significant effects, using a generous $p < 0.5$ cutoff and keeping species to absorb non-phylogenetic effects before testing for phylogenetic effects[77]. In the final model, we calculated partial and total $R^2$ values using *R2_lik* in the *rr2* package[78]. We used the Yekutieli-Benjamini-Hochberg procedure[79] to control the false discovery rate when making multiple tests, assuming conservatively that $2 \times 53$ comparisons are either independent or positively correlated.

To summarize the results of phylogenetic regression, we tallied the numbers of traits which responded to site $P/E_p$ and species $P/E_p$ as predicted. For cases where responses were consistent with predictions for all species except *dumosa* and sometimes *arenacea* (the species with the two lowest values of species $P/E_p$), we discounted tallies from 1 to 0.9 or 0.8. For cases where responses were consistent with predictions for all gardens except Hattah and sometimes Bealiba (the two driest gardens), we discounted tallies to 0.75 or 0.5.

**Principal components analysis.** We used phylogenetically unstructured PCA to analyze multivariate responses to site and species $P/E_p$, to address the fact that many traits are closely correlated with each other and provide a contrast to the regression analyses that take phylogeny into account. No phylogenetically structured PCA model yet exists to deal with multiple species scored at multiple sites. All 53 traits included in the regression analyses were centered and standardized for inclusion in the PCA and then site $P/E_p$ and species $P/E_p$ were projected onto the PCA. We performed the PCA with the *prcomp* function (*stats* package[80]). We regressed PC1 and PC2 against site $P/E_p$, species $P/E_p$, and their difference. Regression against site $P/E_p$−species $P/E_p$ effectively allows PC1 to be related to species $P/E_p$ across species within

individual gardens, and to be related to site $P/E_p$ within species across gardens.

**Adaptive crossover.** We assessed species performance by comparing species survival, final height, final shoot mass, realized height H* (survival*final height), and realized mass M* (survival*final mass) within and across gardens. H* and M* take dead plants into account, effectively treating them as having zero live height or shoot mass. Sample sizes differed for survival (all plants), height (all surviving plants) and mass (a subset of surviving plants). Therefore, calculation of H* and M* standard errors must consider this. We calculated s.e. from:

$$\text{s.e.}^2 = \left( \hat{p}^2 + \frac{\hat{p}(1-\hat{p})}{n'} \right) \frac{s^2}{n} + m^2 \frac{\hat{p}(1-\hat{p})}{n'} \tag{1}$$

where $\hat{p}$ is the portion of surviving plants, $n'$ is the number planted, and $n$ is the number of plants from which the trait mean ($m$) and standard deviation ($s$) were measured. To compare H* and M*, for each species pair, we calculated $p$ values from z-scores, assuming a normal distribution. We then applied a Benjamini-Hochberg correction to $p$ values within each site. We compared mean survival in a similar manner, adding Yates' continuity correction. We compared H and M within each site using Tukey's Honest Significant Differences (HSD) test.

Comparing mean ± s.e. values of individual performance indices for species within a garden allows inferences regarding significant differences among species there. The large sample size for survival and height (81–254 individuals initially) allows resolution of much finer differences between species in % survival and final height than for final mass.

### Reporting summary

Further information on research design is available in the Nature Portfolio Reporting Summary linked to this article.

## Data availability

The data generated in this study and presented in figures are available as Source data.

## Code availability

The R scripts created to perform key analyses in this study are available from Zenodo (https://doi.org/10.5281/zenodo.8277056).

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

## Acknowledgements

This study was supported by NSF IOS-1557906 awarded to T.J.G., K.A.M., and M.A.A., and by a grant to K.A.M. from the Wisconsin Alumni Research Foundation. Swinburne University provided substantial logistical support; University of Melbourne provided access to lab and office space. We wish to thank Alexandra Barlow, Jessica Bolden, Luke Gebert, Sachinthani Karaunarathne, Markus Löw, Mathias Neumann, and Simon Parsons for field assistance, and Morgan Bakken, Daryan Fisher, Sophia Gosetti, Kennah Konrad, Cecilia Vanden Heuvel, and Sophia Ward for help with anatomical measurements. UW students aided in excavating roots, supported in part by the Humboldt Fund of the Department of Botany. Sarah Friedrich helped draft several of the figures. David Ackerly and two anonymous reviewers provided helpful comments. Special thanks to Michael Kemp for assistance and support throughout. Permits for our common gardens were kindly issued by DELWP Victoria (permit NW11041F).

## Author contributions

D.D.S. conducted the field study, assisted by M.A.A., A.M.S., C.K., K.A.M., and T.J.G. T.J.G., K.A.M., and M.A.A. designed and supervised the study. D.D.S. and C.A. analyzed the data; T.J.G. and D.D.S. wrote the paper, with input from all other authors.

## Competing interests

The authors declare no competing interests.
