## [Peer Review File · Nature Communications]

Ecophysiological adaptations shape distributions of closely related trees along a climatic moisture gradientREVIEWER COMMENTS

Reviewer #1 (Remarks to the Author):

This paper describes a truly impressive long term (5 years) common garden experiment involving 10 species of eucalypt from contrasting home-site climates grown together in each of 4 common gardens situated along a strong moisture gradient (P/Ep) in Victoria, Australia. The authors make measurements of 50 growth and survival traits, including a wide range of allometric, physiological, hydraulic, and economic traits throughout and at the end of the experiment. Given the experimental design, the study asks whether these 50 traits differ in a predictable way (according to optimality theory) with moisture availability measured across species distribution as evidence for adaptive and genetic differences. It asks whether these traits differ in a predictable way within species between sites as evidence of plasticity, and whether these differences are stronger than the genetic differences. Finally, it asks whether species home-site advantage is demonstrated as a function of fitness in terms of increased height and survival compared to non-local species.

Strong associations with both species P/Ep and site P/Ep were detected for most traits providing evidence of genetic differentiation and plasticity in a predictable way. However, there were some notable exceptions of traits that did not fit the authors expectations, some of which could be explained biologically. PCA revealed that genetic associations were stronger than differences in plasticity. Finally, home-site advantage and adaptive cross-over was revealed only when height was measured relative to survival. Overall, while the authors observed some strong patterns in their data, the many exceptions to the rule diminishes their strength and raises questions regarding generalisability.

The paper is mostly well written, although at times there is a brevity in writing that does not allow concepts to be fully explored. The methods are mostly clearly described, although I have some concerns (see below) and the analysis seems to be robust especially placed within a phylogenetic framework. The results section was at times not clearly presented, and the inclusion of the many exceptions to the expected pattern were somewhat confusing and diminishing. The Discussion is somewhat convoluted, with much of the text dedicated to the various exceptions to the rules and the possible circumstances that may have led to them occurring. Also, the focus on specific traits such as MPS is not well grounded in the study, including the Introduction.

The Introduction does not provide a strong grounding for the reader to think about why the many traits that were measured should vary in a predictable way in response to changes in moisture availability. There is a strong emphasis on optimality theory, but this idea is not well meshed with the later sections on leaf-level physiological and specific trait responses in the discussion. The authors rely on a qualitative assessment of the various trait data in relation to species and site P/Ep. The list of traits and their expected response to changes in moisture availability seems simplistic. Perhaps there is scope to frame these trait responses within the findings of previous studies on eucalyptus.

The focus of this analysis is on moisture availability as a driver of adaptation. But the sites and species distributions also vary in other climate characteristics especially temperature. Furthermore, temperature is likely to have a strong effect on some of the traits of interest. Did the authors consider the effect of temperature in their study design?

Along these lines, the authors mention in the Introduction possible interactions with fire along the moisture gradient in selecting particular traits. But these interactions are not considered formally in the analysis or discussed qualitatively. Of course, there are complex influences of a wide range of abiotic and biotic drivers of trait selection along environmental gradients and focusing on moisture helps constrain the study parameters. But even so, the overall picture of the traits and how they vary within species across gardens and across species within gardens is not clearly presented, possibly on account of the vast number of traits studied and the differences in their level of adaptation or not, as well as the number exceptions that diminish the generality of the rule.

Specific comments:

Ln110-117: This section is not easy to understand possibly due to an overload of information in a

single long sentence with various examples of where exceptions to the rule were observed.

Ln118: Nmass did not increase with site P/Ep according to Fig 2.

Ln166: This description of climate within the study period highlights the need to assess whether the climate experienced at the sites during 5 years of the experiment are typical and agree with long term data.

Figure 2: Why are the relevant cells only partially thickened in grey? What does the grading from red to blue signify?

Figure 3: It is clear enough to see how traits vary with site P/Ep, including overall trends. However, the regression lines for individual species in many instances don't seem to fit the data (e.g. SLA for *E. viminalis*). Also, why are the curves for each species the same shape?

Figure 4: The nominal categorisation of traits into fixed, plastic, or both is potentially useful in identifying traits that are adaptive or maladaptive. However, it is very hard to digest all the information presented. The level of trait coordination, if present, is not clearly detectable. Also, what's the differences between traits that show an association with site P/Epm (in blue) and those that show an association with site P/Ep but not species P/Ep (salmon)? Presumably, the former also does not show an association with species P/Ep. Please clarify this. Also, there is no discussion or indeed mention of these results in the Discussion, so I'm left wondering about its significance. In plot b, I'd suggest separating into two plots the fitted lines for site and species, respectively.

Methods:

A description of how predawn and midday water potentials were measured is not provided, including measurement frequency and timing throughout the experiment. This is important considering that changes in water potential through time and differences in water potential between sites directly relates to moisture availability – especially predawn as an estimate of soil water potential. Expanding this information may help develop a picture of how the plants of the different species in the different gardens responded through time.

It is not clear whether the sequential points used to measure g_{min} were linear. It is the linear phase following stomatal closure of the mass loss over time relationship that is used to calculate g_{min} . Please clarify.

Reviewer #2 (Remarks to the Author):

This manuscript reports on an intensive experimental study of 10 Eucalyptus species grown in 4 common gardens, with measurement of numerous physiological and functional traits and fitness traits including survival, biomass, and height growth. The analyses are presented as tests of the predictions of optimality theory for both plastic and genetic variation in traits across gradients, as well as tests of local adaptation based on adaptive cross-over.

Studies of multiple taxa in multiple common gardens are extremely valuable, and we need more of them to provide tests of adaptive variation and plasticity. This paper covers a comprehensive range of traits with clear and succinct analyses addressing variation within and among species. The results provide strong support for adaptive responses in this system, with some intriguing cases where variation across site environment and species environment differ.

I have two overall concerns about the paper in its present form. First, the introduction presents the study as a test of 'theory' regarding trait variation, and 'the usually untested optimality assumption', for trait responses to environment and resulting fitness variation. I find these overstatements. First, the predicted responses for the many traits are drawn from a combination of theoretical and empirical literature (citations 1-32). In some cases these are theoretical predictions, but often they are based on previously documented empirical patterns. I appreciate

the paragraph starting on p83. As clarified here, the predictions for the traits are qualitative (i.e. the direction of the effect) and address whether the patterns 'appear to be adaptive'. This is a much weaker claim than saying that the results are tests of theory - to a large extent these patterns have been tested previously, and this study is a particularly comprehensive and confirmatory analysis for this study system.

The second point is the distinction between adaptive cross over (or what I would call local adaptation), vs. tests of optimality. Showing that a species does better in its home environment compared to other species is a strong test of adaptation. But it is not a test of optimality, and certainly not a test that individual traits take on optimal values. Moreover, and perhaps more important, adaptive cross over tests are most clear cut when they involve a species or population in 'home' vs. 'away' environments. However, in this case there are ten species and just 4 environments, and the driest garden is outside the environmental range for any of the species. As a result, the analysis of adaptive cross over becomes largely descriptive (L143-L180), and frankly quite difficult to follow without very close examination of the figures in relation to the text.

As an additional point, I did not find the interpretation of Fig. 3 clearcut. Since some traits have negative and other positive predicted relationships, how does one know which are aligned in adaptive vs. maladaptive directions. root.m/shoot.m is in the pink sector, implying higher values are negative on PCA2. This is aligned with species P/Ep, and I would expect high values of root/shoot to be found in drier conditions. So isn't this a maladaptive fixed trait. Moreover according to Fig. 2, this trait does not have a significant relationship with either factor, and it is not indicated to be counter to theory for the relationship with site P/Ep. So the PCA figure is not consistent with Fig. 2, and it seems to imply results that are not shown by individual analyses.

For a discussion of traits which show different qualitative responses to environmental vs. genetic/adaptive gradients, the concepts of cogradient and countergradient variation may provide a helpful framework (see for example Lusk et al. 2008 Trends in Ecology and Evolution)

Specific comments:

L56 Thicker bark is expected for fire survival or resprouting but is not found in obligate seeding species that evolve in fire prone environments. This is an example of a trait that does not have a single predicted relationship, as it depends on the life history context of the species.

L69 Is P/Ep based on annual totals for each factor? Does this account for seasonality of rainfall or evaporative demand (i.e. the difference between summer rain and summer dry - mediterranean - climates)

L105-110 what are these different percentages?

Fig. 1 - It would be helpful to show the Victoria boundary on species maps to align with the climate map

Fig. 2. Since there are separate tests for the environmental and species relationships, what does the single r^2 value refer to?

Fig 4, last sentence of legend: 'species P/Ep' is repeated - one of the should say 'site'?

In sum, I find this a comprehensive and insightful analysis which will be an extremely valuable contribution to the functional ecology literature.

This review is signed
David Ackerly

Reviewer #3 (Remarks to the Author):

General appreciation:

I have read the manuscript ID NCOMMS-23-1106 entitled "Adaptation drives tree distributions along a climatic moisture gradient." The authors used an impressive dataset on plant traits across a moisture gradient and found little contradictions with the general expectations of trait-environmental associations at the site level. They also showed that adaptation might be playing a critical role in driving species distributions across environmental gradients.

The study presents a good amount of work. The authors compile novel data. Their analyses are extensive and properly executed. I must say that I enjoyed reading their manuscript; in my view, it is a strong paper. This paper will be well received by evolutionary biologists and ecologists. The results are intriguing, though, as outlined below, I found myself wanting to see more of the theoretical predictions and a deep discussion of the implications of the results presented.

Comments:

1) The introduction is well-written, and the authors set a series of expectations but fail to make clear what are the expected association given their data and/or the underlying theory. The introduction should be more specific about the predictions/expectations given their data. For example, the authors stated, "...by asking if the observed relationships of a trait to relative moisture supply in their home ranges (as measured by species P/Ep averaged over each species' range) accord qualitatively with economic theory." [P5, L85-87], but no direction of the expected relationship is presented. I suggest the authors add a schematic figure or table to explain the expected associations between traits as a function of moisture supply.

By doing this, the authors will provide the readers a clear guidance on how to interpret their results. This is just a suggestion, but I believe the work's quality can improve.

2) Phylogenetic regressions. The authors used a phylogenetic hypothesis published elsewhere and used phylogenetic multilevel analyses to test the variation in traits across moisture supply while accounting for different group-levels (species, sites). This is a relatively novel implementation in eco-evolutionary studies; however, although including phylogenetic information can improve model predictions and interpretability, its inclusion is not always a good option, especially when the number of species is very low, as is the case of this study which evaluates ten species. In other words, with increasing taxonomic sampling, the associated error in the estimates decreases and the reverse.

Although the authors presented an extensive analysis of phylogenetic regressions, I suggest the authors to perform comparisons between models with and without phylogenetic information and check if the phylogenetic information is contributing to the overall estimates or simply adding noise. I want to stress that this is not a small issue, and it must be addressed to ensure the quality of the conclusions; if not, then my objection is wrong, and you can ignore me, though please explain why.

3) The discussion section is interesting; however, the authors discuss very little about the role of the evolutionary mechanisms driving the reported patterns. In lines 276-283, they discuss about the rapid adaptation in *Eucalyptus*, but no further discussion of why this happens is provided. In other words, yes, evolutionary adaptations matter, but better discussion and literature review on this issue should be included.

I hope the authors don't find my comments/suggestions offensive or aggressive, but I hope at least they find these comments helpful.

REVIEWER COMMENTS

We'd like to thank the reviewers for their many thoughtful and useful comments. Below, we have numbered each reviewer comment for clarity to which we respond in blue.

Reviewer 1

1. This paper describes a truly impressive long term (5 years) common garden experiment involving 10 species of eucalypt from contrasting home-site climates grown together in each of 4 common gardens situated along a strong moisture gradient (P/Ep) in Victoria, Australia. The authors make measurements of 50 growth and survival traits, including a wide range of allometric, physiological, hydraulic, and economic traits throughout and at the end of the experiment. Given the experimental design, the study asks whether these 50 traits differ in a predictable way (according to optimality theory) with moisture availability measured across species distribution as evidence for adaptive and genetic differences. It asks whether these traits differs in a predictable way within species between sites as evidence of plasticity, and whether these differences are stronger than the genetic differences. Finally, it asks whether species home-site advantage is demonstrated as a function of fitness in terms of increased height and survival compared to non-local species.

Strong associations with both species P/Ep and site P/Ep were detected for most traits providing evidence of genetic differentiation and plasticity in a predictable way. However, there were some notable exceptions of traits that did not fit the authors expectations, some of which could be explained biologically. PCA revealed that genetic associations were stronger than differences in plasticity. Finally, home-site advantage and adaptive cross-over was revealed only when height was measured relative to survival.

Many thanks to the reviewer for this enthusiastic response!

2. Overall, while the authors observed some strong patterns in their data, the many exceptions to the rule diminishes their strength and raises questions regarding generalisability.

Only six of $2 \times 53 = 106$ trait-environment relationships overall show a hard contradiction to the predictions being tested, and five of these apparent exceptions (stomatal conductance and related traits) are explicable in terms of an *a priori* model proposed by Givnish (1986) and points to a fundamental shortcoming in the widely cited Farquhar-Cowan model. The Givnish model is based on the increase in mesophyll photosynthetic sensitivity – MPS, the decline in photosynthetic capacity with drops in water potential, after the effects of stomatal conductance are removed – in species from drier climates (lower P/Ep) which we recently documented in the *Eucalyptus* species studied in the current paper (Salvi et al. 2021, 2022). Another eight traits are consistent with predictions except for the driest site or species with the driest distributions. Thus, the few exceptions are either explained on *a priori* grounds (and are therefore not exceptions) or have a pattern that unifies them. We make these points at the end of the first section of the Results (lines 132-140) to provide clarity and avoid confusion. Thanks to the reviewer for raising this issue.

3. The paper is mostly well written, although at times there is a brevity in writing that does not allow concepts to be fully explored. The methods are mostly clearly described, although I

have some concerns (see below) and the analysis seems to be robust especially placed within a phylogenetic framework. The results section was at times not clearly presented, and the inclusion of the many exceptions to the expected pattern were somewhat confusing and diminishing. The Discussion is somewhat convoluted, with much of the text dedicated to the various exceptions to the rules and the possible circumstances that may have led to them occurring.

We appreciate the reviewer's suggestion that some points should be explained at greater length, so that brevity does not cloud clarity. We have thus added small amounts of text to the **Abstract** (lines 2, 5, 13, 21), **Introduction** (lines 35, 38-42, 59-60, 63-65, 82-89), **Results** (117, 119, 122, 131-140, 145-151, 154-163, 167-168, 209-215), **Discussion** (lines 227-232, 238-239, 251, 259-260, 305-307, 315-316, 323-342, 351-371), and **Supplemental Information** to (a) clarify the central role of the economics of gas exchange in shaping most of the traits under consideration, (b) explain with a few examples why traits should vary with relative moisture supply, (c) outline the economics tradeoffs behind the predicted responses of almost all traits being studied; and (d) emphasize the traits that confirm the *a priori* predictions and thus provide context for the relatively few exceptions (see response to #2 above).

4. Also, the focus on specific traits such as MPS is not well grounded in the study, including the Introduction.

The Introduction does not provide a strong grounding for the reader to think about why the many traits that were measured should vary in a predictable way in response to changes in moisture availability. There is a strong emphasis on optimality theory, but this idea is not well meshed with the later sections on leaf-level physiological and specific trait responses in the discussion. The authors rely on a qualitative assessment of the various trait data in relation to species and site P/E_p. The list of traits and their expected response to changes in moisture availability seems simplistic. Perhaps there is scope to frame these trait responses within the findings of previous studies on eucalyptus.

As noted under the previous point, we have inserted text to explain further how optimality theory can lead to predicted trends in several specific traits as a function of moisture supply relative to evaporative demand. **To address the reviewer's concerns regarding hypothesis specificity, we have added a new column to Table S1 (formerly part of Table S3) to outline the tradeoff(s) affecting each trait and the expected shift in optimal expression toward moister sites.**

MPS is **not** a trait studied in the common gardens of the present study. We discuss it in detail where it is first appropriate, in the Discussion, to explain the apparently paradoxical behavior of stomatal conductance and related traits. But we also mention it in functional terms *without naming it* in the critical second paragraph of the Introduction (lines 30-31) and then foreshadow its application in the Discussion with lines 133-134 in the Results.

We state in the Abstract and Introduction that we evaluated predictions in qualitative terms – whether a trait increases or decreases with site P/E_p and species P/E_p. The reality in ecology is that the predictions of optimality models are almost always evaluated in **qualitative** terms – whether a trait increases or decreases as one or more environmental parameters vary. This is generally true whether we are considering such different approaches as *optimal*

foraging theory (only a few studies have quantified handling times, processing costs, and caloric or other benefits to produce quantitative predictions of diet), *optimal breeding systems* (monoecy, dioecy, or hermaphroditism as a function of flower size and mechanism of seed dispersal), or *optimal stomatal conductance* (all tests to date have been semi-quantitative). In this paper, we evaluate predictions for a large number of traits, so that even tests of qualitative predictions can yield high significance using signs tests.

Our documentation of how many traits vary **quantitatively** as a function of both species and site P/E_p is itself ground-breaking and significant for future efforts in many areas – for example, in calibrating earth-systems models and allowing traits values and their plasticity to be parameterized as a function of climatic conditions. We now make this point in the Discussion.

Unfortunately, there is no analogue to our multi-species, multi-garden, multi-trait study in previous studies of *Eucalyptus* with which to make comparisons.

5. The focus of this analysis is on moisture availability as a driver of adaptation. But the sites and species distributions also vary in other climate characteristics especially temperature. Furthermore, temperature is likely to have a strong effect on some of the traits of interest. Did the authors consider the effect of temperature in their study design?

Good question, calling for important clarifications. We chose our study transect because it shows extensive variation across sites in climatic moisture supply (precipitation P), evaporative demand (E_p), and moisture supply relative to demand (P/E_p) but relatively little variation in mean annual temperature, latitude, and rainfall seasonality. **We combined historical temperature and precipitation data to produce new Fig S1 depicting Walter-Lieth climate diagrams showing seasonality at each site.** Our sites show little seasonality in rainfall and little change in rainfall seasonality as total rainfall increases (see Figure S1). P/E_p varies nearly 8-fold among our sites, corresponding to the large environmental variation between temperate forests and semi-deserts, but MAT varies by only 5.7°C from 16.9°C in Hattah to 11.2°C in Toolangi. This difference is unlikely to have a big direct effect but should have a strong indirect effect through its impact on pan evaporation E_p (together with the effects of cloudiness and humidity, both strongly related to precipitation P) and thus on relative moisture supply P/E_p . Variation in P/E_p is very strongly correlated ($r > 0.999$) with P^2 alone. Based on species and site values, P , E_p , and MAT are all strongly correlated with each other ($r^2 > 0.83-0.99$), so all three are intertwined in their effects on P/E_p . Temperature variation along our transect is unlikely to have major effects except through its impact on P/E_p , and we are unaware of models predicting optimal values of any trait under study based on temperature alone vs. its indirect effect on evaporative demand. **We make these points on lines 82-89 of the text.**

6. Along these lines, the authors mention in the Introduction possible interactions with fire along the moisture gradient in selecting particular traits. But these interactions are not considered formally in the analysis or discussed qualitatively. Of course, there a complex influences of a wide range of abiotic and biotic drivers of trait selection along environmental gradients and focusing on moisture helps constrain the study parameters. But even so, the overall picture of the traits and how they vary within species across gardens and across species within gardens is not clearly presented, possibly on account of the vast number of traits studied and the differences in their level of adaptation or not, as well as the number

exceptions that diminish the generality of the rule.

We argue that drier conditions favor more fires that should, in turn, favor thicker and denser bark in resprouting trees, and more stems per plant. Moister conditions should favor fewer fires, favoring denser plant cover and thus plants with growth forms resulting in greater height for a given stem mass. Those four predictions relate bark thickness, bark density, stem number, and stem allometry to P/E_p . There is no “interaction” which we predicted (or now envision) between *independent* effects of P/E_p and fire, and we did test all four predictions formally. In any case, we conducted no controlled burns at our study sites to test for such interactions.

We disagree with the statement that a clear picture is not presented of how traits vary within species across gardens and across species within gardens. Graphs showing exactly how all 53 traits vary within species as a function of site P/E_p across gardens, as a function of species P/E_p within gardens, and as two-dimensional responses to both site and species P/E_p are provided in the text for selected traits (Fig 3) and in the Supplemental Information for all traits (Fig S2). Model fits and data points are displayed. In response to reviewer 3, we have also added a column to Figure 2 giving the explicit prediction for each trait; the rationale for each of these predictions is given in Table S1 (formerly part of Table S3). We also provide equations and values of r^2 and P (Fig S2, Table S5). As noted above, the reviewer’s view regarding the number of exceptions to the predictions does not agree with our findings. The number of actual exceptions is small, and five of the six “hard” exceptions are well explained in the context of the cited Givnish model incorporating mesophyll photosynthetic sensitivity to the economics of gas exchange.

7. Specific comments:

Ln110-117: This section is not easy to understand possibly due to an overload of information in a single long sentence with various examples of where exceptions to the rule were observed.

Good point. We’ve rewritten this section to simplify, increase clarity, and address the closely related point already raised by the reviewer (point 2 above) – see lines 122-140.

8. Ln118: N_{mass} did not increase with site P/E_p according to Fig 2.

Good catch. We’ve rewritten the sentence (line 131) and noted that the decline in N_{mass} with site P/E_p seen in Fig. 3 is not significant, as we had already correctly noted in Fig. 2.

9. Ln166: This description of climate within the study period highlights the need to assess whether the climate experienced at the sites during 5 years of the experiment are typical and agree with long term data.

Important point. We now compare monthly and annual observed of temperature and rainfall against historical climatic data for the 1970-2000 reference period. These data are shown in **new supplemental Figs S9 and S10**. Our sites maintained climate differences during the study period like those in the reference period. In 2018 and 2019, precipitation was below average and temperatures were above average at the two drier sites (Bealiba and Hattah). In 2020 and 2021, temperatures were closer to average. Site ranks were switched in 2020 when Toolangi was drier and warmer than Mt Disappointment. Data at Toolangi were incomplete in 2021 due to weather station vandalism. **These points are now made in the**

Discussion (lines 209-215) and the captions of Figures S9 and S10.

10. Figure 2: Why are the relevant cells only partially thickened in grey? What does the grading from red to blue signify?

The caption for Figure 2 explains these two conventions.

11. Figure 3: It is clear enough to see how traits vary with site P/E_p , including overall trends. However, the regression lines for individual species in many instances don't seem to fit the data (e.g., SLA for *E. viminalis*). Also, why are the curves for each species the same shape?

Model curves are often the same shape for each species (or site) because there was no significant interaction between the effects of site P/E_p and species P/E_p . For this interaction to be significant, slopes must differ in proportion to species (or site) P/E_p , which is not often the case. Similarly, the form of the fixed effects portion of the model (shown in Fig. 3) constrains the fit such that intercept differences are proportional to P/E_p differences. Indeed *E. viminalis* has higher SLA than predicted from its native P/E_p . We also considered random effects in each model, which allows further variation in intercepts due to site, species, phylogeny and phylogenetic attraction. In the case of SLA, species as a random effect was significant ($p < 0.001$) which would allow the model to fit *viminalis* better. We chose to only show the fixed effects predictions of the models for clarity.

12. Figure 4: The nominal categorisation of traits into fixed, plastic, or both is potentially useful in identifying traits that are adaptive or maladaptive. However, it is very hard to digest all the information presented. The level of trait coordination, if present, is not clearly detectable. Also, what's the differences between traits that show an association with site P/E_{pm} (in blue) and those that show an association with site P/E_p but not species P/E_p (salmon)? Presumably, the former also does not show an association with species P/E_p . Please clarify this. Also, there is no discussion or indeed mention of these results in the Discussion, so I'm left wondering about its significance. In plot b, I'd suggest separating into two plots the fitted lines for site and species, respectively.

Thank you for pointing out that the information was difficult to digest, as presented. We made several clarifications and noted several changes below.

First, the level of trait-trait coordination can be inferred from the degree to which the vectors corresponding to those traits in the 2-dimensional PCA are parallel (positive correlation), anti-parallel (negative correlation), or orthogonal (no correlation or coordination). **We have added this point to the text (lines 145-163).** Formal analyses of trait-trait relationships would double this paper's length; we will present those results elsewhere.

Second, **we clarified the text and figure legend** to indicate that the degree of trait-environment coordination can be inferred from the degree to which the trait vector is parallel, anti-parallel, or orthogonal to the vectors for species P/E_p (reflecting the native moisture environment of species) and site P/E_p (reflecting the moisture environment of individual gardens). **See lines 145-163, especially last two lines.**

Third, we have **substantially modified Figure 4** to indicate more clearly whether the observed response of each trait to P/E_p in each phylogenetic regression agrees or disagrees with predictions. Specifically, vector color now indicates the strength of

agreement/disagreement with expectations based on species P/E_p , and the terminal dot indicates the same regarding expectations based on site P/E_p . See Figure 4 caption. We removed colors indicating trait category ("hydraulic", "allocational" etc.) to reduce the number of color palettes. We had made no mention in the text of these categories or their (lack of) pattern.

Fourth, based on preceding changes, traits in the blue sectors are strongly correlated (positively in the NE sector, negatively in the SW sector) with site P/E_p , and thus increase (or decrease) strongly as a plastic response to site (garden) P/E_p . This expectation – and the strength of trait associations with site or species P/E_p – are based on angular position BUT can be confirmed/evaluated by examining the colors of the vectors and terminal dots for each trait in each sector. Because the blue sectors are orthogonal to the light-green sectors, blue traits show little or no correlation with species P/E_p ... and thus, with fixed differences associated with species distributions in nature. Traits in the salmon sector are less strongly positively correlated with site P/E_p and are negatively correlated with species P/E_p . Red vectors for most of the traits in the salmon sector indicate that the traits there mostly disagree with predicted responses to species P/E_p . Specifically, most show the expected plastic response to increased moisture supply across gardens (e.g., stomatal conductance increases toward moister gardens) but the opposite trend across species within gardens (e.g., stomatal conductance is higher in species from drier climates within gardens). Hence, we labeled the salmon sector as “maladaptive” fixed traits. We use quotation marks here because this superficial maladaptation is explicable when invoking mesophyll photosynthetic sensitivity. **In response to the reviewer’s suggestion, we added a brief comment on the significance of this graph in the Results (lines 157-163).**

Sixth, we adopted the reviewer’s second suggestion and **separated plot b into two panels** (now b and c) to show the response of PCA axis 1 score against site and species P/E_p separately. Doing this has the added advantage of making the shallower slopes of the plastic vs. fixed responses to site and species P/E_s values visually more obvious; plastic responses are less marked (shallower slopes) than fixed responses.

13. Methods:

A description of how predawn and midday water potentials were measured is not provided, including measurement frequency and timing throughout the experiment. This is important considering that changes in water potential through time and differences in water potential between sites directly relates to moisture availability – especially predawn as an estimate of soil water potential. Expanding this information may help develop a picture of how the plants of the different species in the different gardens responded through time.

Thanks for pointing out this oversight. We now state that predawn and midday water potential were measured during the first harvest and how they were measured. Several factors hindered a strong characterization of water potential over time, including equipment theft, COVID lockdowns, and the time and expense of visiting sites on a climatic gradient 430 km long. See lines 683-685.

14. It is not clear whether the sequential points used to measure g_{min} were linear. It is the linear phase following stomatal closure of the mass loss over time relationship that is used to calculate g_{min} . Please clarify.

Agreed. In brief terms, the points chosen for g_{min} came from (1) when mass vs time was quasi-linear and (2) after an initial high-conductance phase. **We now state in the Methods that mass typically declined rapidly and then achieved a quasi-linear relationship to time. We state that minimum conductance came from this phase.** On a more technical level, g indeed typically starts high and then declines to some lower level. This level varies in its stability. Conductance may be very stable or may decline gradually over time. In the long term, conductance should reach zero as the leaf comes into equilibration with the atmosphere. But at that point the leaf will have died. This combination of sometimes unstable conductance and eventual tissue death prompted us to find where conductance reached its minimum but while leaf tissue appeared healthy, as assessed by F_v/F_m measurements. See lines 664-668.

In our initial submission, our calculation of leaf diffusive conductance had a small error. After recalculation, the values of g_{min} have increased but the overall pattern of the regression of g_{min} against site P/E_p and species P/E_p is unchanged. Although the pattern did not change, the fit improved and r^2 increased from 0.56 to 0.67 (see Figs 2 and S2). Corrections have been incorporated in all the analyses, including the PCA, with essentially no change in the summary Figure 4.

Reviewer 2

1. This manuscript reports on an intensive experimental study of 10 Eucalyptus species grown in 4 common gardens, with measurement of numerous physiological and functional traits and fitness traits including survival, biomass, and height growth. The analyses are presented as tests of the predictions of optimality theory for both plastic and genetic variation in traits across gradients, as well as tests of local adaptation based on adaptive cross-over.

Studies of multiple taxa in multiple common gardens are extremely valuable, and we need more of them to provide tests of adaptive variation and plasticity. This paper covers a comprehensive range of traits with clear and succinct analyses addressing variation within and among species. The results provide strong support for adaptive responses in this system, with some intriguing cases where variation across site environment and species environment differ.

Many thanks to the reviewer for this enthusiastic response!

2. I have two overall concerns about the paper in its present form. First, the introduction presents the study as a test of 'theory' regarding trait variation, and 'the usually untested optimality assumption', for trait responses to environment and resulting fitness variation. I find these overstatements. First, the predicted responses for the many traits are drawn from a combination of theoretical and empirical literature (citations 1-32). In some cases, these are theoretical predictions, but often they are based on previously documented empirical patterns. I appreciate the paragraph starting on p83. As clarified here, the predictions for the traits are qualitative (i.e., the direction of the effect) and address whether the patterns 'appear to be adaptive'. This is a much weaker claim than saying that the results are tests of theory - to a large extent these patterns have been tested previously, and this study is a particularly comprehensive and confirmatory analysis for this study system.

Tests of qualitative predictions are nevertheless tests of theory. Furthermore, all but

one of the previous tests of theory to which the reviewer refers were themselves qualitative tests, hinging on slopes up or down, not attempts to evaluate their quantitative values. Even tests of the Cowan-Farquhar model for optimal stomatal conductance g ($\partial A(g,t)/\partial E(g,t) = \lambda$) have only been semi-qualitative, because λ was a fit parameter rather than one based on independently quantifying the biological and physical bases for the Lagrangian multiplier λ .

We agree with the reviewer that our study “is a particularly comprehensive analysis for this study system”. We would add that the large number of hypotheses we test qualitatively – largely formulated on the unified basis of the economics of gas exchange – allows an overall quantitative test (signs test) of models based on that principle. That “meta” test is different in nature from the qualitative tests of individual models in the past. Quantitative fits of regressions to predictor variables – when neither the slope or intercept of those regressions is predicted – are, ultimately, each only a qualitative test of theory as well, evaluating whether the trait in question rises, falls, or shows no simple response to the putative driver.

We stated explicitly in the abstract that “Phylogenetically structured tests show that most trait-environment relationships accord qualitatively with theory.” We characterized the nature of our claims immediately and did not in any way hide the limitations of our approach. Those limitations were explained in the early paragraph beginning on original ms. line 83.

We are sorry if we in any way contributed to the reviewer’s impression that “only in some cases these are theoretical predictions, but often they are based on previously documented empirical patterns”. For all but one trait (leaf C content), we are testing optimality models or inferences based on previously published tradeoffs. To make that clear, and address concerns raised by both reviewer 1 and 2 – we have added a column to Table S1 (formerly part of Table S3) in the Supplementary Information giving the tradeoffs/rationale involved.

For the reasons given above, we do not feel that we are overstating the case by saying that we are testing an unprecedented number of theoretical predictions as to how plant traits related to energy capture should vary along a gradient of moisture supply. Some of the patterns involved (e.g., increase in leaf width with rainfall) have been known since at least von Humboldt, but only in the last few years or decades have compelling arguments been advanced to account for these patterns, producing predictions that can be tested in many contexts, some unexpected. The responses of stomatal conductance (and functionally related traits) to species P/E_p within gardens was completely unexpected based on previous observations and detailed physiological studies but was predicted by a previous model (Givnish 1986).

3. The second point is the distinction between adaptive cross over (or what I would call local adaptation), vs. tests of optimality. Showing that a species does better in its home environment compared to other species is a strong test of adaptation. But it is not a test of optimality, and certainly not a test that individual traits take on optimal values. Moreover, and perhaps more important, adaptive cross over tests are most clear cut when they involve a species or population in 'home' vs. 'away' environments. However, in this case there are ten species and just 4 environments, and the driest garden is outside the environmental range for any of the species. As a result, the analysis of adaptive cross over becomes largely descriptive (L143-L180), and frankly quite difficult to follow without very close examination of the figures in relation to the text.

We understand the distinctions that the reviewer outlines, and agree with most, but disagree with a few of his conclusions. **First**, we agree that “showing that a species does better in its home environment compared to other species is a strong test of adaptation”. But given how we measured “better”, it is indeed also a test of the underlying assumption of optimality models – species with a particular set of predicted traits grow faster in height than others under the conditions the former species dominate. We stand by our conclusion on this matter, and on the central point that only rarely are such context-specific advantages in growth ever demonstrated. As the reviewer knows, some ecological models – indeed, whole schools of thought – assume that no such context-specific growth advantage exist and that they are **not** the basis for specific distributions along environmental gradient. J. P. Grimes’ widely cited CSR theory, for example, is based on precisely the latter assumptions, and argues that in unproductive environments the ecological dominants should NOT be the species with a local growth advantage. We have added this point in the Discussion (see lines 328-335).

Second, we wholeheartedly agree with the reviewer that our study does not test whether individual traits (e.g., g , SLA, [N]) take on optimal values. To capture this point and the one we made in the last paragraph, we have inserted statements in the Discussion regarding the significance of our results regarding the optimality assumption AND the inability of our analysis to test whether species are adopting optimal values of traits (lines 335-338). This meshes with our statements (and earlier conclusions by Givnish and Montgomery) that not all patterns of plasticity are adaptive or quantitatively optimal. We argue that the limits of species distributions are based in part on the failure of individual taxa to possess optimal traits or combinations of traits in many environments. Failure to acquire such optimal traits via plasticity (or local genetic adaptation) in traits affecting energy capture should thus play a key role in setting species distributions (see lines 338-342). • **Both points are key, and we thank reviewer 2 for pushing us to articulate our rationale and conclusions better.** • By the way, we do plan to **calculate** the optimal value of particular traits (e.g., g_s) from observed values of others (e.g., mesophyll photosynthetic sensitivity, root hydraulic conductance, A vs. c_i curves), and compare those calculated values with observations in another ms.

Third, we agree and disagree with the reviewer regarding tests of adaptive cross-over:

- (a) The reality of the distributions of individual *Eucalyptus* species regionally (and often, within stands) is that those distributions often overlap – as we illustrate for our study species in Figure 1D. We adopted this system not only to study it for its own merits but as a model system for understanding different but overlapping distributions of dominant species along environmental gradients. Any study showing that species A has an advantage in environment E_A , B in E_B , C in E_C , ... and N in E_N almost surely is not examining species distributed along a gradient (where many species have differing but overlapping ranges), and almost surely is not studying systems in which multiple dominants coexist. We believe our system accurately represents the complexity of species distributions along our study gradient and many other environmental gradients.
- (b) In our system, it would not be enough to have even 10 common gardens in which to score traits, growth, and survival in 10 *Eucalyptus* species. Ideally, we should have

run such gardens at many more sites (20, 30, 50?) along the continuous P/E_p gradient. But that was and is impossible in practical terms. With a multi-year study involving 10 species x 4 sites and a budget of nearly \$1M, we were hard-pressed to complete the studies we did and obtain an unprecedented set of results.

- (c) It is incorrect to claim that “the driest garden is outside the environmental range for any of the species”. As shown in Figure 1D, Hattah is within the P/E_p ranges of four species. *Eucalyptus dumosa* grows naturally in the site's immediate vicinity. However, we did (and do) acknowledge that the study would have been more powerful if we had been able to add a fifth site at P/E_p between 0.65-0.8, bridging the gap between Bealiba and Mt. Disappointment where a number of our species reached peak abundance. We were not able to locate and permit a site there. A positive way to look at this is that none of these species whose abundance peaks between 0.6-0.8 P/E_p had a growth advantage at the four existing sites ... as expected. **We have added a short statement to that effect in the text – see lines 305-307.**
- (d) We agree with the reviewer’s statement that Figure 5 is hard to follow, given that it requires integration with other figures, especially Figure 1. It was also unnecessarily complex because it plots survival, height growth, mass growth, realized height growth, and realized mass growth vs. site P/E_p for each of 10 study species. **To remedy this situation**, we have done three things. **First**, we have eliminated all but one panel from Figure 5, focusing solely on the relationship of realized height growth to site P/E_p – which is a focus in the text. **Second**, we have added color-coded bars to the top of the figure indicating where on the P/E_p axis individual *Eucalyptus* species are the first or second most abundant species regionally, based on Figure 1D. This should make it easier to see how the pattern of cross-over is, in fact, adaptive. **Third**, we have made similar changes (bars) to all the panels in the original Figure 5 and added it as a figure in the Supplemental Information (Fig S5). That will allow readers to confirm our statement that the metrics other than realized height growth do not show adaptive cross-over. That supplemental figure should now be easier to interpret because we’ve made the learning curve less steep, simplifying and clarifying the central panel to become the new version of Figure 5. Readers will naturally look at the new comprehensive figure in the SI after they’ve absorbed the lessons and conventions of Figure 5.

4. As an additional point, I did not find the interpretation of Fig. 3 clearcut. Since some traits have negative and other positive predicted relationships, how does one know which are aligned in adaptive vs. maladaptive directions. Root.m/shoot.m is in the pink sector, implying higher values are negative on PCA2. This is aligned with species P/E_p, and I would expect high values of root/shoot to be found in drier conditions. So isn't this a maladaptive fixed trait. Moreover, according to Fig. 2, this trait does not have a significant relationship with either factor, and it is not indicated to be counter to theory for the relationship with site P/E_p. So, the PCA figure is not consistent with Fig. 2, and it seems to imply results that are not shown by individual analyses.

Good points. **First**, in the Figure 4 caption (admittedly, deep in the caption) we stated that “Red vectors indicate responses counter to expectations in phylogenetic regressions.”

Second, the reviewer's comments on root mass/shoot mass identify a problem with the relationship between Figure 2 (summary of phylogenetic regressions) and Figure 4. Indeed, that trait does not have a significant relationship to site or species P/E_p but does have significant interaction between site and species P/E_p , thus showing a response that runs one way for the dry site(s) and another for the moister sites. **Third**, to clarify in the PCA whether the regressions of traits were or were not significantly related to site P/E_p (or species P/E_p), we have now color-coded the vector and terminal dot for each trait to indicate whether its regression agrees or disagrees with the expected trend with species P/E_p (vector body) and site P/E_p (terminal dot). This should allow readers to a) evaluate whether vectors in different descriptive (not prescriptive) sectors of the PCA agree or disagree with expected responses to site or species P/E_p and b) see the strength of those responses in the phylogenetic regressions. See also our response to reviewer 1, point 12 (see lines 145-163 and Fig. 4 caption). **Fourth**, all readers need to recognize that PCA captures linear/ monotonic responses, not diatonic or more complex responses; nonlinear responses to P/E_p are best judged by the graphs of regressions we provide for each trait (see lines 556-558). **Fifth**, the new coding of vectors should make clear that traits in different sectors do *generally* agree or disagree with the predicted responses to site and species P/E_p . As indicated in the caption and lines 158-162, the "maladaptive plasticity" sector is based on angular placement in the PCA, but contains an odd mix of three traits, only one of which the phylogenetic regressions identify as maladaptive plasticity.

5. For a discussion of traits which show different qualitative responses to environmental vs. genetic/adaptive gradients, the concepts of cogradient and countergradient variation may provide a helpful framework (see for example Lusk et al. 2008 Trends in Ecology and Evolution).

Thanks for the head's-up on this; we have inserted the phrase and citation (lines 238-239). There is, alas, no simple rule for the traits that show co- vs. countergradient variation. We provide a direct rationale for why maximizing energy capture should result in countergradient variation in g_s and related traits.

6. Specific comments:

L56 Thicker bark is expected for fire survival or resprouting but is not found in obligate seeding species that evolve in fire prone environments. This is an example of a trait that does not have a single predicted relationship, as it depends on the life history context of the species.

Good point. We have added the conditional "in perennial resprouting trees (and thin bark in obligate seeders)" to the phrase "Adaptation to fire, herbivores or pathogens, and tradeoffs involving allocation to unproductive support tissue should favor relatively thicker and denser bark³³", as well as citing Schwilk & Ackerly 2001 re obligate seeders and thin bark (lines 59-60). Among our study species, only *Eucalyptus regnans*, *nitens*, and *viminalis* are (near-)obligate seeders, surviving ground fires but dying after high-intensity fires that open the canopy and permit seedling establishment. These three species – native to the moistest regions – would be favored to have thin bark based on either the usual argument for perennial, resprouting trees or the argument for reseeders; see mention on line 60.

7. L69 Is P/E_p based on annual totals for each factor? Does this account for seasonality of rainfall or evaporative demand (i.e., the difference between summer rain and summer dry -

Mediterranean - climates)

P/E_p comes from the annual means of precipitation and pan evaporation. Please also see our response to reviewer 1, point 5, and lines 82-89 in the text, regarding the minimal seasonality of precipitation. We have added Walter-Lieth climate diagrams (Fig S1) showing seasonality in temperature and rainfall at each site. Our study sites show little seasonality in rainfall, and little change in rainfall seasonality, as total rainfall increases. Moisture availability – as indicated on the climate diagrams by the difference between the curves for mean monthly rainfall and temperature – is somewhat seasonal, with a modest drop in midsummer. None of our sites exhibit Mediterranean climate, as all receive summer rainfall.

8. L105-110 what are these different percentages?

We have rewritten this section (lines 116-118) to clarify that the percentages refer to the fraction of trait relationships to site or species P/E_p that are qualitatively consistent with predictions, whether the regressions of those traits on P/E_p are significant or not. Later in the same paragraph, and in Figure 2, we give the fraction of traits showing significant relationships to site or species P/E_p that accord with predictions.

9. Fig. 1 - It would be helpful to show the Victoria boundary on species maps to align with the climate map.

Agreed. We now show the border of Victoria with each species distribution.

10. Fig. 2. Since there are separate tests for the environmental and species relationships, what does the single r^2 value refer to?

Good catch. As now noted in the figure caption, the single value of R^2 refers to the multiple regression using only the fixed values of site and species P/E_p .

11. Fig 4, last sentence of legend: 'species P/E_p ' is repeated - one of the should say 'site'?

Agreed and corrected – thanks.

12. In sum, I find this a comprehensive and insightful analysis which will be an extremely valuable contribution to the functional ecology literature.

This review is signed
David Ackerly
dackerly@berkeley.edu

Thanks, David, for your uniformly insightful and helpful comments.

Reviewer 3

1. General appreciation:

I have read the manuscript ID NCOMMS-23-1106 entitled “Adaptation drives tree distributions along a climatic moisture gradient.” The authors used an impressive dataset on plant traits across a moisture gradient and found little contradictions with the general

expectations of trait-environmental associations at the site level. They also showed that adaptation might be playing a critical role in driving species distributions across environmental gradients.

The study presents a good amount of work. The authors compile novel data. Their analyses are extensive and properly executed. I must say that I enjoyed reading their manuscript; in my view, it is a strong paper. This paper will be well received by evolutionary biologists and ecologists. The results are intriguing, though, as outlined below, I found myself wanting to see more of the theoretical predictions and a deep discussion of the implications of the results presented.

Many thanks to the reviewer for this enthusiastic response!

2. Comments:

1) The introduction is well-written, and the authors set a series of expectations but fail to make clear what are the expected association given their data and/or the underlying theory. The introduction should be more specific about the predictions/expectations given their data. For example, the authors stated, "...by asking if the observed relationships of a trait to relative moisture supply in their home ranges (as measured by species P/E_p averaged over each species' range) accord qualitatively with economic theory." [P5, L85-87], but no direction of the expected relationship is presented. I suggest the authors add a schematic figure or table to explain the expected associations between traits as a function of moisture supply.

By doing this, the authors will provide the readers a clear guidance on how to interpret their results. This is just a suggestion, but I believe the work's quality can improve.

In response to the reviewer's suggestion, we have added a column to Figure 2 giving the predicted trend in each trait and indicated that in the caption. Given the original coding of Figure 2, those predictions were there, but they were implied rather than explicit. As indicated in the caption, Figure 2 used/uses color in other columns to indicate the observed direction of the trend, and thickened borders on cells to indicate a significant disagreement from the prediction. We additionally revamped part of Table S3 (now Table S1) to detail all predictions. We have used the reviewer's idea to indicate unambiguously the direction of the proposed relationships of each trait to species and site P/E_p . We have also revised Figure 4 (PCA) to make agreement/disagreement of traits with expected relationships to site and species P/E_p more obvious. **Thanks!**

3. 2) Phylogenetic regressions. The authors used a phylogenetic hypothesis published elsewhere and used phylogenetic multilevel analyses to test the variation in traits across moisture supply while accounting for different group-levels (species, sites). This is a relatively novel implementation in eco-evolutionary studies; however, although including phylogenetic information can improve model predictions and interpretability, its inclusion is not always a good option, especially when the number of species is very low, as is the case of this study which evaluates ten species. In other words, with increasing taxonomic sampling, the associated error in the estimates decreases and the reverse.

Although the authors presented an extensive analysis of phylogenetic regressions, I suggest the authors to perform comparisons between models with and without phylogenetic information and check if the phylogenetic information is contributing to the overall estimates or simply adding noise. I want to stress that this is not a small issue, and it must be addressed to ensure the quality of the conclusions; if not, then my objection is wrong, and you can ignore me, though please explain why.

Excellent suggestion. We retained the phylogenetic regression approach in the text and figures, but we also redid all the regressions using a standard, phylogenetically unstructured approach, tabulated the results in the same style as Fig 2 (see Fig S4). We report in the text (line 138-140) that the unstructured analyses give very nearly the same picture as the phylogenetically structured analyses (compare tables at foot of Figs 2 and S4).

4. 3) The discussion section is interesting; however, the authors discuss very little about the role of the evolutionary mechanisms driving the reported patterns. In lines 276-283, they discuss about the rapid adaptation in Eucalyptus, but no further discussion of why this happens is provided. In other words, yes, evolutionary adaptations matter, but better discussion and literature review on this issue should be included.

Agreed. We did not originally include such a discussion before based solely on word limits for Nature Communication papers. We recommend inserting the following paragraph starting on line 352-372:

Rapid divergence in climatic distributions and associated adaptations in eucalypt species corresponds to extreme aridification in Australia starting 3 Mya, following a long period of drying and increased seasonality 25-10 Mya^{44,45}. *Eucalyptus* began diversifying 52 Mya under warm wet conditions but did not become dominant until sclerophyll vegetation spread under drier conditions starting 25 Mya; most species did not evolve until arid vegetation appeared in the Plio-Pleistocene⁴⁶. The more arid-adapted subgenus *Symphomyrtus* – including 79% of *Eucalyptus* species and the great majority of those in mallee – began rapid species diversification ~5 Mya, with the most rapid diversifications in the last million years⁴⁶. Selection should have favored traits adapted to drier conditions in species evolving in drier areas, with ecological sorting of existing species – based on context- and trait-dependent survival and competitive ability – then shaping the distributions of species with different traits along gradients of P/E_p . Genomic scans of populations of *Eucalyptus tricarpa* (related to *E. sideroxylon*) along aridity gradients in southeastern Australia suggest that adaptations to moisture supply have occurred across the genome: 73 of 94 loci showing significant deviations among sites also have significant correlations with site P/E_p ⁴⁷. Evidence for selection across the genome associated with relative moisture supply – and correlations with specific traits and genes – should now be sought across species with different distributions along the climatic moisture gradient.

5. I hope the authors don't find my comments/suggestions offensive or aggressive, but I hope at least they find these comments helpful.

Thanks for each of your suggestions!

REVIEWERS' COMMENTS

Reviewer #1 (Remarks to the Author):

The authors comprehensively respond to each of the reviewers' comments from the first round of peer review. Associated changes and text additions have improved clarity and strengthened the manuscript. This study represents a huge amount of work with insights that advance our understanding of plant adaptation across gradients of aridity and lead to better models of how plants will respond to changes in climate.

Reviewer #2 (Remarks to the Author):

I am reviewing a revision of this paper, and reiterate my earlier comments about the value and unusual nature of this study - very few studies span so many sites, species, and traits. The detailed, trait-specific results will provide extensive material for the physiological ecology community, perhaps of most interest where a priori expectations weren't met or there were significant interaction effects (shown only in supp. material).

I appreciate the revision made in response to previous reviews, but reiterate some of my previous concerns and, now on a second reading (sometimes required to fully appreciate a paper), I have some that did not catch my attention previously. Many of my comments below identify a common theme of overstating the novelty of the paper (i.e. 'one of the first', L373). Such statements usually only serve to annoy readers who are well versed on the topic.

1. Most importantly, I still maintain that the focus on 'optimality' and statements about 'maximizing growth' are overused. An early statement that ecophysiological theory and empirical study provide extensive qualitative predictions of the direction of response is appropriate and the study tests these qualitative predictions. Not all the theory is optimality theory per se (i.e. a theory of the direction of adaptive response is different than a theory that attempts to model or solve for the optimal value of a trait), and many of the citations underlying the predictions are of empirical studies. This does not weaken the study.

2. The paper is very strongly grounded in the ecophysiological literature, but less so in the literature of local adaptation and adaptive crossovers. This is an extensive research field, though stronger among students of ecotypic variation within species, rather than variation among species. The now fairly old review by Kawecki and Ebert (<https://doi.org/10.1111/j.1461-0248.2004.00684.x>) is foundational, and a forward citation search will reveal many relevant papers. This recent review appears useful: <https://besjournals.onlinelibrary.wiley.com/doi/full/10.1111/1365-2745.13695>. For trees, the studies of ecotypic variation in *Pinus contorta* are among the best - this paper and many others: [https://doi.org/10.1890/0012-9615\(1999\)069\[0375:GRTCIP\]2.0.CO;2](https://doi.org/10.1890/0012-9615(1999)069[0375:GRTCIP]2.0.CO;2)

3. L373 This is an unacceptable overstatement. The entire literature of the relationship between traits and distributions addresses this linkage, though the majority of studies don't also address plasticity (but this statement doesn't say anything about plasticity). Two examples, but the list is very long: <https://bsapubs.onlinelibrary.wiley.com/doi/10.2307/2656722>, <https://doi.org/10.1073/pnas.2008987118>. Moreover, this study does not implicate the role of specific traits, as stated on L378. The study shows that the entire complex of adaptive traits is apparently related to adaptive crossover, but the role of any one trait cannot be isolated. With apologies for pointing to one of my own papers, this old review on the challenges of identifying the adaptive role of individual traits may be useful: [https://doi.org/10.1641/0006-3568\(2000\)050\[0979:TEOPET\]2.0.CO;2](https://doi.org/10.1641/0006-3568(2000)050[0979:TEOPET]2.0.CO;2)

4. There are two critical decisions that appear to be made post-hoc, which then lead to the claims of the strong support of the data for theory. Of most concern, the authors have 5 fitness traits: survival, height, mass, Height*, and Mass* (Fig. S5). They then state (L293) that only H* exhibits adaptive crossover, and that is the one shown in the paper. In the Discussion, the result is used as

evidence that height is therefore presumably the most important for fitness. This interpretation is self-reinforcing. Either one knows a priori which fitness trait is most important, and then can test adaptive crossover. Or one assumes adaptive crossover, and then uses this assumption (as done here) to determine which trait is apparently favored by selection. But with five traits to choose from, it's not a strong test to choose, after the fact, which one to use, because it fits theory. I actually don't think the authors are wrong about the importance of height for trees, and the authors should investigate independent literature on this topic, rather than relying only on the internal logic.

The second decision is the ad-hoc treatment of H^* for *E. viminalis* at Bealiba. While the explanation in the text is somewhat reasonable, one must accept that data is data, and the standards for ad-hoc adjustments of a value (and this is a large adjustment) should be high, especially when it so conveniently upholds the a priori predictions being tested. The expectation at Bealiba is a quadratic with a peak in the middle - did you test a quadratic? Another approach would be to take the functions fit to the data in the first two rows of Fig. 6, and multiply them together; multiplying a positive and a negative relationship (above) will give a negative quadratic.

5. Why is the model by Givnish(5) only introduced in the Discussion to explain unexpected results. It's an old paper, so shouldn't it be included among the sources used in the introduction to set up the predictions, and if there are contrasting predictions among different papers in the literature, then Fig. 2 could be modified to reflect this and the results would not be counted as 'disagree', but rather as helping to resolve contrasting predictions.

6. Finally (L141) I find this PCA very difficult to interpret, and would suggest removing it entirely. PCA is not well suited to interpreting alignment of individual vectors (i.e. the degree to which vectors align, in the first two axes, may be weakly related to the pairwise relationship between the traits). If the goal is to classify traits into categories, that is better accomplished with Fig. 2, using the combination of sig/nonsig, and whether responses follow the predicted direction, to put into the different categories. PCA biplots usually show both the row and column positions, i.e. using vectors for the traits, and then having the 40 points in the background, showing the species-site combinations. I find the regressions for PCA axis scores more useful. If you keep this figure, then I would remove the quadrant analysis, which is quite confusing, and focus on interpretation of axis 1 scores.

Specific comments:

Fig 2 - explain half boxes, and pink/blue shaded boxes. And I recommend adding a symbol, perhaps along the right side, of which traits had significant interaction terms.

L117 Fig. 2 lists 26 significant responses in both columns, which would be <50%. How does this align with text? Also, L117-119 vs. L120-122: Is the difference in these percentages based on whether responses are significant. Clarify - most readers will assume that only significant results are reported and worthy of reporting, so perhaps flip and first state significant responses and then how many additional, but non-sig, were also in the predicted direction.

Fig. 3 - Since the model used $\log(P/E_p)$, why not show that on the x-axis, so the regression lines are linear. The curves are otherwise a bit confusing until one gets deep into the methods and sees they are due to log-transforming environment. Also, can you indicate which panels show relationships that are opposite to expectation (mirroring the information provided in Fig. 2)

Fig. 6

- What is the function used for survival at Hattah? Did you test non-linear models for the other relationships, or make this decision based on visual inspection of the data?
- The ad-hoc exclusion of arenacea at Hattah also gives the feeling of post-hoc decisions to obtain cleaner statistical relationships, rather than excepting that nature throws exceptions at us!

L238 The brief mention of countergradient variation is appreciated. Not all theory leads to parallel predictions for interspecific variation and variation due to plasticity. If it's not apparent, I do believe that evolution is very effective, and most variation in nature is adaptive. So when we are surprised, we either need a conceptual framework to understand the constraints that will produce

non-adaptive variation, or our theory is inadequate. I think countergradient results are among the most interesting, as they do require a deeper conceptual framework. It's worth noting that with leaf physiology, there are some key differences in predictions between deciduous and evergreen species, relevant here. Regarding contrasting effects of moisture supply, vs. evaporative demand, see (with another apology for self-citation):

<https://nph.onlinelibrary.wiley.com/doi/full/10.1111/j.1469-8137.2007.02208.x>

L330 Is this actually a contradiction with CSR, when you include survival. I.e. your results showing H^* higher at low resources was not for height per se, but height*survival, and CSR theory also focuses on the tolerance and ability to survive, not just growth, in low resource environments. Perhaps Tilman's related tolerance vs. competitiveness framework is more useful here, rather than CSR.

L686 Do you know what vessel lengths are in these species? Many researchers in the hydraulics community believe that open vessels lead to errors in K_{stem} , and that segments need to be longer than the longest vessels for these methods to work.

This review is signed
David Ackerly

RESPONSES TO VERBATIM REVIEWER COMMENTS

Reviewer 1

1. The authors comprehensively respond to each of the reviewers' comments from the first round of peer review. Associated changes and text additions have improved clarity and strengthened the manuscript. This study represents a huge amount of work with insights that advance our understanding of plant adaptation across gradients of aridity and lead to better models of how plants will respond to changes in climate.

Many thanks for this supportive evaluation!

Reviewer 2

1. I am reviewing a revision of this paper, and reiterate my earlier comments about the value and unusual nature of this study - very few studies span so many sites, species, and traits. The detailed, trait-specific results will provide extensive material for the physiological ecology community, perhaps of most interest where a priori expectations weren't met or there were significant interaction effects (shown only in supp. material).

Thank you for this enthusiastic evaluation!

I appreciate the revision made in response to previous reviews, but reiterate some of my previous concerns and, now on a second reading (sometimes required to fully appreciate a paper), I have some that did not catch my attention previously. Many of my comments below identify a common theme of overstating the novelty of the paper (i.e. 'one of the first', L373). Such statements usually only serve to annoy readers who are well versed on the topic.

2. Most importantly, I still maintain that the focus on 'optimality' and statements about 'maximizing growth' are overused. An early statement that ecophysiological theory and empirical study provide extensive qualitative predictions of the direction of response is appropriate and the study tests these qualitative predictions. Not all the theory is optimality theory per se (i.e. a theory of the direction of adaptive response is different than a theory that attempts to model or solve for the optimal value of a trait), and many of the citations underlying the predictions are of empirical studies. This does not weaken the study.

Our views do not differ much from those of the reviewer and we believe that his and our interpretations are wholly compatible. Specifically, we believe his concern is that we are not demonstrating that the observed trait values are optimal values – that is, the values that would maximize the realized rate of height growth in particular environments. We share that view. But we are not claiming to demonstrate this. We only state that our findings are consistent with the qualitative predictions of large numbers of optimality models. All theories advanced are based on optimality theory as outlined and documented with citations in Supplemental Data 2 (a supplemental table in the previous version of the manuscript). As we stated in response to the reviewer's concern when he first raised it, many tests of optimality models in ecology use the same approach

we have. We are planning follow-up papers that will test whether particular traits (e.g., stomatal conductance, leaf:root allocation) are, in fact, close to calculated optima.

3. The paper is very strongly grounded in the ecophysiological literature, but less so in the literature of local adaptation and adaptive crossovers. This is an extensive research field, though stronger among students of ecotypic variation within species, rather than variation among species. The now fairly old review by Kawecki and Ebert (<https://doi.org/10.1111/j.1461-0248.2004.00684.x>) is foundational, and a forward citation search will reveal many relevant papers. This recent review appears useful: <https://besjournals.onlinelibrary.wiley.com/doi/full/10.1111/1365-2745.13695>. For trees, the studies of ecotypic variation in *Pinus contorta* are among the best - this paper and many others: [https://doi.org/10.1890/0012-9615\(1999\)069\[0375:GRTCIP\]2.0.CO;2](https://doi.org/10.1890/0012-9615(1999)069[0375:GRTCIP]2.0.CO;2)

Agreed. We now cite several studies of local adaptation within species on lines 105-110. Our paper focuses on adaptive crossover among species, and on the failure of plasticity to result in each species showing the same maximization of realized height growth at each site. These are topics not addressed by these studies mentioned, which is why we did not initially cite them. We now cite two studies that are the only multi-species analogues that we know of.

4. L373 This is an unacceptable overstatement. The entire literature of the relationship between traits and distributions addresses this linkage, though the majority of studies don't also address plasticity (but this statement doesn't say anything about plasticity). Two examples, but the list is very long: <https://bsapubs.onlinelibrary.wiley.com/doi/10.2307/2656722>, <https://doi.org/10.1073/pnas.2008987118>. Moreover, this study does not implicate the role of specific traits, as stated on L378. The study shows that the entire complex of adaptive traits is apparently related to adaptive crossover, but the role of any one trait cannot be isolated. With apologies for pointing to one of my own papers, this old review on the challenges of identifying the adaptive role of individual traits may be useful: [https://doi.org/10.1641/0006-3568\(2000\)050\[0979:TEOPET\]2.0.CO;2](https://doi.org/10.1641/0006-3568(2000)050[0979:TEOPET]2.0.CO;2)

Agreed. We never intended readers to see that sentence in isolation – **only** in combination with the later sentences. We originally conditioned the sentence by including a summary of our novel methodological contributions, but then removed those conditions because they were detailed in the later sentences. Which led to the unacceptable sentence. Our rewrite of the concluding statement – on lines 431-460 – discusses conditions, approaches, and significance, and removes explicit statements re priority.

5. There are two critical decisions that appear to be made post-hoc, which then lead to the claims of the strong support of the data for theory. Of most concern, the authors have 5 fitness traits: survival, height, mass, Height*, and Mass* (Fig. S5). They then state (L293) that only H* exhibits adaptive crossover, and that is the one shown in the paper. In the Discussion, the result is used as evidence that height is therefore presumably the most important for fitness. This interpretation is self-reinforcing. Either one knows a priori which fitness trait is most important, and then can test adaptive crossover. Or one assumes adaptive crossover, and then uses this assumption (as done here) to determine which trait is apparently favored by selection. But with five traits to choose from, it's not a strong test to choose, after the fact, which one to use, because it fits theory. I actually don't think the authors are wrong about the importance of height for trees, and the authors should

investigate independent literature on this topic, rather than relying only on the internal logic.

We respectfully disagree that this was a post-hoc decision. This study was founded with the vision that competition and natural selection in a given context should favor *Eucalyptus* species that maximize realized height growth there, based on (a) shade intolerance of the *Eucalyptus* species in question (as stated in the original ms), and (b) the need to consider both growth and survival in estimating likely success (see Givnish et al. 2004 and its discussion, as well as Kobe et al. 1995). That was our hypothesis from the outset, and we present trends in realized height growth with P/E_p and compare the trends for survival, height growth, mass growth, and realized mass growth to drive home the point that maximizing realized height growth is favored at each P/E_p and not the other metrics. We now cite Kobe et al. 1995, Givnish et al. 2004, and Charles et al. 2018 (lines 391-3) to emphasize that realized height growth incorporates two key components of fitness in trees that help determine distributions along environmental gradients; the last citation of Charles et al. 2018 supports the conclusion (with which reviewer Ackerly concurs) that relative height gives an advantage in further height growth.

6. The second decision is the ad-hoc treatment of H^* for *E. viminalis* at Bealiba. While the explanation in the text is somewhat reasonable, one must accept that data is data, and the standards for ad-hoc adjustments of a value (and this is a large adjustment) should be high, especially when it so conveniently upholds the a priori predictions being tested. The expectation at Bealiba is a quadratic with a peak in the middle - did you test a quadratic? Another approach would be to take the functions fit to the data in the first two rows of Fig. 6, and multiply them together; multiplying a positive and a negative relationship (above) will give a negative quadratic.

Convenient or not, there is a clear ecological and statistical rationale for our ad hoc treatment of H^* for *E. viminalis* at Bealiba. In response to the reviewer's comment, we calculated a quadratic fit for H^* as a function of species P/E_p at Bealiba without correcting H^* for *E. viminalis* and found that it does in fact peak close to the site P/E_p at Bealiba, as expected. A similar quadratic fit for H^* at Hattah shows a much smaller departure from the linear fit shown in the paper; similar fits for H^* at Mt. Disappointment and Toolangi show no visible departure for the linear fits in the paper. • We appreciate the suggestion regarding the multiplication of survival of the separately estimated functions of survival and height growth, but also see the possibility of deviations of individual species from those trends, which might cause their actual realized height (calculated from their actual height growth and survival) to deviate from the expectations based on the across-species trends. We thus believe it is better to look – as we did – at differences in realized height growth as a function of P/E_p than at the product of the across-species functions of survival and height growth.

7. Why is the model by Givnish(5) only introduced in the Discussion to explain unexpected results. It's an old paper, so shouldn't it be included among the sources used in the introduction to set up the predictions, and if there are contrasting predictions among different papers in the literature, then Fig. 2 could be modified to reflect this and the results would not be counted as 'disagree', but rather as helping to resolve contrasting predictions.

Good question. If you were to go to an ESA or ASPB meeting and ask 100 people whether they expect – based on predictive models – that stomatal conductance should increase with moisture supply, we believe that 97-100% would say yes. The Givnish 1986 model has been out there 37

years, but almost everyone in the field is only familiar with/only cites Cowan and Farquhar 1977 and derivative models. So increased stomatal conductance with moisture supply is the community expectation. Rhetorically, we therefore believe it is more effective to highlight our surprising countergradient findings re stomatal conductance and related traits by showing in the Discussion how those findings can be explained by the Givnish 1986 model. Delaying that model – which is only in the last few years being cited by large numbers of papers – to the Discussion does lead to an overestimate of the apparent deviations from predictions, but the rhetorical impact is greater. If we advanced the Givnish 1986 model as *the* model to account for stomatal conductance in the Intro, a lot of people would immediately object in their minds; even if we subsequently justify the choice of model in the Discussion, it will take up the same space, and require the same confrontation of models. So, we much prefer the sequence used. We have, however, inserted a note in the Intro re the Givnish 1986 model, so that it does not come as a complete surprise in the Discussion. We had previously mentioned that model in former Table S1 (now Supplemental Data 1) and continue to do so.

8. Finally (L141) I find this PCA very difficult to interpret, and would suggest removing it entirely. PCA is not well suited to interpreting alignment of individual vectors (i.e. the degree to which vectors align, in the first two axes, may be weakly related to the pairwise relationship between the traits). If the goal is to classify traits into categories, that is better accomplished with Fig. 2, using the combination of sig/nonsig, and whether responses follow the predicted direction, to put into the different categories. PCA biplots usually show both the row and column positions, i.e. using vectors for the traits, and then having the 40 points in the background, showing the species-site combinations. I find the regressions for PCA axis scores more useful. If you keep this figure, then I would remove the quadrant analysis, which is quite confusing, and focus on interpretation of axis 1 scores.

We understand your suggestion but respectfully disagree. Figure 2 and the 2- and 3-dimensional regressions in Figs S2 provide one way of classifying traits into categories, but PCA provides a different statistical approach. The advantages of displaying the PCA results are that (1) it shows that the rather different approaches yield similar results, so that they are robust, and (2) it shows graphically (rather than by reference to tables) how trait responses are related to each other, to site P/E_p , and to species P/E_p . We strongly believe that the PCA figure is an extremely important and original contribution, and that, with reflection, should be understandable by wide swaths of the Nature Communications readership. Indeed, PCA biplots often show the vectors and the points, but you will understand that adding all site*species points would crowd an already full figure. In our opinion, correlations between PC1 and P/E_p (i.e. Fig 4bc) were more useful to show, given the space.

Specific comments:

9. Fig 2 - explain half boxes, and pink/blue shaded boxes. And I recommend adding a symbol, perhaps along the right side, of which traits had significant interaction terms.

We agree and have added the explanation and column showing interaction significance.

10. L117 Fig. 2 lists 26 significant responses in both columns, which would be <50%. How does

this align with text? Also, L117-119 vs. L120-122: Is the difference in these percentages based on whether responses are significant. Clarify - most readers will assume that only significant results are reported and worthy of reporting, so perhaps flip and first state significant responses and then how many additional, but non-sig, were also in the predicted direction.

We retained the current order to save space but have modified this reporting to clarify that we are first considering all relationships (regardless of significance) and that we are using weighting to account for traits that agreed for most but not all sites or species, and mention the tallies in Supplemental Data 2. This should make it clearer where these percentages are coming from and why we are not simply reporting 26/53 (49%) as the fraction of traits that fully, significantly agree.

11. Fig. 3 - Since the model used $\log(P/E_p)$, why not show that on the x-axis, so the regression lines are linear. The curves are otherwise a bit confusing until one gets deep into the methods and sees they are due to log-transforming environment. Also, can you indicate which panels show relationships that are opposite to expectation (mirroring the information provided in Fig. 2).

We now mention the log transformation in the caption, and believe that should remove the confusion that the reviewer highlights. We believe plotting an arithmetic, untransformed axis is easier for readers to grasp than log-transformed axis scaling and certainly easier than plotting data with log-transformed values.

12. Fig. 6

- What is the function used for survival at Hattah? Did you test non-linear models for the other relationships, or make this decision based on visual inspection of the data?

- The ad-hoc exclusion of *arenacea* at Hattah also gives the feeling of post-hoc decisions to obtain cleaner statistical relationships, rather than excepting that nature throws exceptions at us!

Thank you for catching this. In the caption we now state the form of the sigmoidal function and that survival at Hattah was the only relationship where this curve was the better fit. Regarding the exclusion of *arenacea*, we see two concise options: (1) we can exclude *arenacea* (as we have done) leading to a very good fit and what should be obvious to the reader that *arenacea* is an outlier to this relationship; or (2) We could keep *arenacea* in the regression, achieve a poor fit for *dumosa*, *arenacea*, *microcarpa* and *sideroxylon*, and leave the reader wondering what is leading to the poor fit. We believe that the first option is better justified and more informative on its own, and it also aided by our discussion for why *arenacea* did not fit well into trends with P/E_p . The exclusion of *E. arenacea* from the survival analysis at Hattah should give the feeling of our having learned something from the experiment and subsequent review of the literature. We ignored soil alkalinity when we chose the ten study species, stratified by their position on the P/E_p gradient, and recognized that almost the entire landscape in the drier part of that gradient (including our Hattah common garden) had sandy soils. We did not, until after the experiment had begun, realize that *E. arenacea* usually grows on acid sands. If we had not excluded *E. arenacea* from the survival regression at Hattah, where alkaline sands predominate, it would have reflected early ignorance compounded by subsequent intransigence to learning from the results and literature. We believe in learning from experience. In any case, we have plotted the outlier survival point and given our rationale for excluding that point.

13. L238 The brief mention of countergradient variation is appreciated. Not all theory leads to parallel predictions for interspecific variation and variation due to plasticity. If it's not apparent, I do believe that evolution is very effective, and most variation in nature is adaptive. So when we are surprised, we either need a conceptual framework to understand the constraints that will produce non-adaptive variation, or our theory is inadequate. I think countergradient results are among the most interesting, as they do require a deeper conceptual framework. It's worth noting that with leaf physiology, there are some key differences in predictions between deciduous and evergreen species, relevant here. Regarding contrasting effects of moisture supply, vs. evaporative demand, see (with another apology for self-citation): <https://nph.onlinelibrary.wiley.com/doi/full/10.1111/j.1469-8137.2007.02208.x>

Agreed.

14. L330 Is this actually a contradiction with CSR, when you include survival. I.e. your results showing H^* higher at low resources was not for height per se, but height*survival, and CSR theory also focuses on the tolerance and ability to survive, not just growth, in low resource environments. Perhaps Tilman's related tolerance vs. competitiveness framework is more useful here, rather than CSR.

We have edited the sentence to state that height growth is not maximized at low P/E_p but realized height growth is, contradicting CSR theory.

We are familiar with the assumptions and predictions of Grime's CSR theory, and our findings on realized height growth contradict CSR theory. There is no place and no argument where Grime predicts adaptive crossover, with growth performance of species native to unproductive habitats outgrowing species from more productive habitats in unproductive conditions. Indeed, Grime 1979 places great emphasis on the fact that – when *Arrhenatherum elatius* (dominant in the most productive British meadows), *Agrostis tenuis* (dominant in mesotrophic meadows), and *Festuca ovina* (dominant in highly unproductive meadows) are grown at high and low levels of nitrate-N supply in a greenhouse – *Arrhenatherum* always has a competitive edge in mixtures (Mahmoud & Grime 1976). This is the very fulcrum of CSR theory. It led Grime to argue that stress-tolerators aren't adapted to outgrowing other plants in unproductive habitats but, instead, to surviving there. Grime uses this theory to argue that evergreen herbs dominate unproductive sites not because they have a growth advantage there, but because they don't have high energy costs. He uses this theory to argue that short leaf stature is favorable on sparsely covered sites not because it results in a growth advantage there, but because they don't have high construction costs to amortize. Givnish 1984 (Am Nat) and 2003 (Silva Fennica) disprove both these claims, showing how evergreens (herbs or woody species) can gain a growth advantage on poor sites while deciduous plants can gain a growth advantage on more fertile (but seasonal) sites, and how short plants can gain an advantage on sparsely covered sites (where there is little net advantage to an increment in leaf height) but a disadvantage on more densely covered sites. Grime's fallacy in claiming that stress-tolerators don't have a growth advantage relative to competitors in unproductive environments is most likely based on his relying on Mahmoud & Grime 1976 and other greenhouse experiments in which not a shred of evidence was presented that the nutrient supply levels used approximated those in the field (that is, in the native environments dominated by different species).

- As we noted under point 5 above, we have included a statement and supporting citations for both

survival and growth rate being important determinants of competitive success in trees.

15. L686 Do you know what vessel lengths are in these species? Many researchers in the hydraulics community believe that open vessels lead to errors in K_{stem} , and that segments need to be longer than the longest vessels for these methods to work.

The open vessel error to which you refer involves the measurement of hydraulic vulnerability – which we did not attempt for the exact reason you give. Maximum hydraulic conductivity (as we have measured) does not require that stem segments be longer than the longest vessels for properly controlled methods to work. We did not measure vessel length due to lack of equipment.

This review is signed
David Ackerly